# Is Bidirectionality Necessary in Mamba for Time Series Forecasting?

## Abstract

Mamba is a sequential model that has recently emerged as a promising alternative to Transformers, offering near-linear complexity. However, although channels in time series (TS) data generally *lack a sequential order*, recent studies have adopted Mamba to capture channel dependencies (CD) in TS, introducing a *sequential order bias*. To address this, prior works have adopted bidirectional Mamba to scan channels in both forward and reverse orders. In this paper, we show that unidirectional Mamba can effectively replace the bidirectional Mamba with simple strategies. To this end, we propose **FSMamba**, a TS forecasting method employing a *unidirectional* Mamba that incorporates a *regularization strategy* to minimize the discrepancy between two embedding vectors generated from data with reversed channel orders, thereby enhancing robustness to channel order. Furthermore, we introduce **channel similarity modeling**, a pretraining task to preserve similarities between channels from the data space to the latent space to enhance the ability to capture CD. Extensive experiments demonstrate the efficacy of our method, achieving state-of-the-art performance on diverse datasets.

## 1 Introduction

Time series (TS) forecasting is prevalent in various fields, including weather (Angryk et al., 2020) and traffic (Cirstea et al., 2022). While Transformers (Vaswani et al., 2017) have been widely employed for this task due to their ability to capture long-term dependencies in sequences (Wen et al., 2022), their quadratic computational complexity limits their application in the real world. Several attempts have been made to reduce the complexity of Transformers (Zhang & Yan, 2023; Zhou et al., 2022); however, they often result in performance degradation (Wang et al., 2025).

To tackle the computational challenges of Transformers, alternatives such as state-space models (SSMs) (Gu et al., 2022) have been considered, employing convolutional operations to process sequences with linear complexity. Recently, Mamba (Gu & Dao, 2023) incorporates a selective mechanism into SSMs to prioritize important information efficiently. Due to its strong balance between performance and efficiency (Wang et al., 2025), Mamba has been widely adopted across various domains (Zhu et al., 2024a; Schiff et al., 2024). In the TS domain, it is utilized to capture temporal dependencies (TD) by processing input TS along the *temporal dimension* (Ahamed & Cheng, 2024), channel

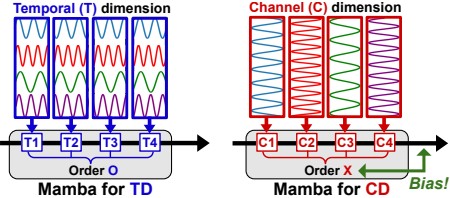

Figure 1: **Sequential order bias.** Capturing CD with Mamba introduces a bias, as Mamba is an SSM-based model designed for *sequential input*, while channels in TS *lack a sequential order*.

dependencies (CD) along the *channel dimension* (Wang et al., 2025), or both (Cai et al., 2024).

It is noteworthy that Mamba is an *SSM-based* model designed for *sequential* inputs, making it more natural to capture TD rather than CD. Nonetheless, we focus on *Mamba capturing CD instead of TD*, following recent works (Liu et al., 2024a; Wang et al., 2025) that adopt complex mechanisms (e.g., Transformer, Mamba) for CD and simpler ones (e.g., MLPs) for TD due to their superior performance. However, directly applying Mamba to capture CD introduces a *sequential order bias* since channels lack an inherent sequential order, as shown in Figure 1.

To address this issue, previous works have employed bidirectional Mamba (Liang et al., 2024; Wang et al., 2025), where two Mambas with different parameters capture CD from a given channel order

| Horizon ($H$) | 96 | 192 | 336 | 720 |
|---|---|---|---|---|
| Bidirectional | **0.139** | 0.165 | **0.177** | 0.214 |
| ① Uni $(1 \rightarrow C)$ | 0.143 | **0.162** | 0.179 | 0.234 |
| ② Uni $(C \rightarrow 1)$ | 0.141 | 0.168 | 0.179 | **0.210** |
| ((① - ②) / ①) | +1.6% | -3.8% | -2.0% | +10.3% |

Table 1: **Limitation of bidirectional Mamba.** 1) *Bidirectional Mamba* may not achieve the best performance, and 2) the performance of *unidirectional Mamba* varies by channel order.

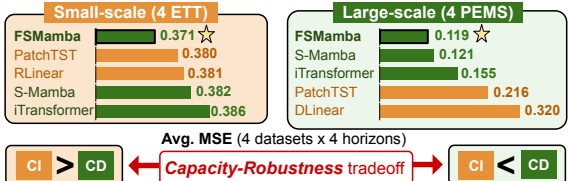

Figure 2: **Performance comparison**. CD and CI models are known to excel on large-scale and small-scale datasets, respectively, and our method outperforms both across all cases.

and its reverse. However, these methods are inefficient due to the need for two models. Another approach involves permuting a channel order during training (Cai et al., 2024) to enhance robustness to the order, while requiring an additional procedure to determine the optimal order for inference.

In this paper, we argue that a *bidirectional Mamba may not effectively address the sequential order bias*, and highlight the need for an effective method to handle the bias. Table 1 shows the performance of the TS forecasting task on the ECL (Wu et al., 2021) dataset using 1) the bidirectional Mamba (Wang et al., 2025) and 2) two unidirectional Mambas with reversed channel orders. The table indicates that 1) the bidirectional Mamba does not consistently achieve optimal performance, and 2) the performance of the unidirectional Mamba varies depending on the channel order.

To this end, we introduce **F**lipped **S**iamese **Mamba** (**FSMamba**), a TS forecasting method employing a *unidirectional* Mamba that handles the sequential order bias by incorporating a *regularization strategy* to minimize the distance between two embedding vectors generated from data with reversed channel orders to enhance robustness to the order. As shown in Table 2, our approach offers a new paradigm for mitigating sequential order bias, providing a more effective and efficient alternative to the conventional bidirectional design. Additionally, we propose **C**hannel **S**imilarity **M**odeling (**CSM**), a pretraining task aimed at improving the model's ability to capture CD by preserving the similarity between channels from the data space to the latent space.

The main contributions of this work are summarized as:

- We propose FSMamba, a TS forecasting method that handles the sequential order bias by regularizing the unidirectional Mamba to minimize the distance between two embedding vectors generated from data with reversed channel orders for robustness to channel order.
- We introduce CSM, a pretraining task that preserves the similarity between channels from the data space to the latent space, enhancing the model's ability to capture CD.
- We conduct extensive experiments with 13 datasets, demonstrating that FSMamba achieves state-of-the-art (SoTA) performance efficiently with unidirectional Mamba. As shown in Figure 2, our method outperforms both CD and channel-independent (CI) models on small and large datasets.

## 2 RELATED WORKS

**TS forecasting with Transformer.** Transformers (Vaswani et al., 2017) are commonly employed for TS forecasting tasks due to their ability to handle long-range dependencies through attention mechanisms. However, their quadratic complexity has led to the development of various methods aimed at improving efficiency, such as modifying the Transformer architecture (Zhang & Yan, 2023; Zhou et al., 2022), patchifying the TS (Nie et al., 2023) or using MLP-based models (Chen et al., 2023; Zeng et al., 2023). While MLP-based models offer simpler structures and reduced complexity compared to Transformers, they tend to be less effective at capturing global dependencies (Wang et al., 2025). Recently, iTransformer (Liu et al., 2024a) inverts the conventional Transformer framework in the TS domain by treating each channel as a token rather than each patch, shifting the focus from capturing TD to CD. This framework has led to significant performance gains and has become widely adopted as the backbone for TS models (Liu et al., 2024b; Dong et al., 2024) to capture CD.

**State-space models.** To overcome the limitations of Transformer- based models, state-space models have been integrated with deep learning to tackle the challenge of long-range dependencies (Rangapuram et al., 2018; Zhang et al., 2023; Zhou et al., 2023). However, these methods are unable to adapt their parameters to varying inputs, which limits their performance. Recently, Mamba (Gu & Dao, 2023) introduces a selective scan mechanism that efficiently filters specific inputs and captures long-range context by incorporating time-varying parameters into the SSM.

**TS forecasting with Mamba.** Due to its balance between performance and computational efficiency, Mamba has also been applied in the TS domain (Xu et al., 2024; Nanbo et al., 2025; Meric Karadag

| | Mamba for TD ($\mathcal{O}(L)$) | | Mamba for TD and/or CD | | | | Mamba for CD ($\mathcal{O}(C)$) | | | |
|---|---|---|---|---|---|---|---|---|---|---|
| | | | **TD or CD** | **TD and CD** ($\mathcal{O}(L \cdot C)$ or $\mathcal{O}(L+C)$) | | | | | | |
| | [A] | [B] | Bi-Mamba+ | SAMBA, MambaMixer MTS-UNMixers | TimeMachine | MambaTS | FMamba | Att.Mamba | S-Mamba | Ours |
| **Attention** | ✗ | O | ✗ | | | | O | | ✗ | |
| **Bias** | ✗ | | O | | | | O | | | |
| **Remove 1D-conv** | ● | ✗ | ✗ | | | ● | ✗ | | | O |
| **Does it handle the bias?** | | | O | | ✗ | O | ✗ | O | | |
| **How to handle the bias?** | | | Bidirectional | | | Permutation | | Bidirectional | | (1) + (2) |

[A]: CMamba, FACTS, ms-Mamba, SiMBA    [B]: SST, Heracles

●: Removing 1D-conv in **Mamba for TD** is *not related to sequential order bias* as it captures *temporal* dependencies (Zeng et al., 2024).

Table 2: **Mamba in TS domain.** While most methods employing Mamba for CD use a bidirectional Mamba to handle the bias, our method adopts two strategies: (1) **Regularization strategy** enables the usage of a *unidirectional* Mamba (instead of a *bidirectional* Mamba) for capturing CD. (2) Unlike previous works which removes 1D-conv from *Mamba for TD*, removing 1D-conv from *Mamba for CD* to handle the bias.

et al., 2025). TimeMachine (Ahamed & Cheng, 2024) utilizes multi-scale quadruple-Mamba to capture either TD alone or both TD and CD. CMamba (Zeng et al., 2024) captures TD with patch-wise Mamba and CD with an MLP and FMamba (Ma et al., 2024) integrates fast-attention with Mamba to capture CD. SST (Xu et al., 2024) employs both Transformer and Mamba to capture local and global TD, respectively. To solve both for vision and time series tasks, Heracles (Patro et al., 2024) combines global/local SSMs with Transformers, while SiMBA (Patro & Agneeswaran, 2024) employs Mamba with frequency-domain channel mixing via Einstein matrix multiplication. SAMBA (Weng et al., 2024), MambaMixer (Behrouz et al., 2024) and MTS-UNMixers (Zhu et al., 2024b) leverage Mamba to capture both CD and TD, decoupling them to reduce computational complexity.

**Bidirectional Mamba for CD.** Recently, various methods (Liang et al., 2024; Weng et al., 2024; Behrouz et al., 2024; Zhu et al., 2024b; Xiong et al., 2025), including S-Mamba (Wang et al., 2025), employ bidirectional Mamba to capture CD by scanning channels in both forward and reverse directions to mitigate sequential order bias. However, they are limited by the need for two independent models (see Figure C.1 for details). MambaTS (Cai et al., 2024) introduces variable permutation training, which shuffles the channel order during training to handle the bias. However, it is limited by the need for an additional procedure to determine the optimal scan order for inference.

## 3 PRELIMINARIES

**State-space model.** SSM transforms continuous input signals $x(t)$ into $y(t)$ via a state representation $h(t)$. This state space represents how the state evolves over time, which can be expressed as:

$$\begin{aligned} h'(t) &= \boldsymbol{A}h(t) + \boldsymbol{B}x(t), \\ y(t) &= \boldsymbol{C}h(t) + \boldsymbol{D}x(t), \end{aligned} \quad (1)$$

where $h'(t) = \frac{dh(t)}{dt}$, and $\boldsymbol{A}, \boldsymbol{B}, \boldsymbol{C}$, and $\boldsymbol{D}$ are learnable parameters. Due to the continuous nature of SSMs, discretization is commonly used to approximate continuous-time representations into discrete-time representations by sampling input signals at fixed intervals, which can be expressed as:

$$\begin{aligned} h_k &= \overline{\boldsymbol{A}}h_{k-1} + \overline{\boldsymbol{B}}x_k, \\ y_k &= \boldsymbol{C}h_k + \boldsymbol{D}x_k, \end{aligned} \quad (2)$$

where $h_k$ and $x_k$ are the state vector and input vector at time $k$, respectively, and $\overline{\boldsymbol{A}} = \exp(\Delta\boldsymbol{A})$ and $\overline{\boldsymbol{B}} = (\Delta\boldsymbol{A})^{-1}(\exp(\Delta\boldsymbol{A}) - \boldsymbol{I}) \cdot \Delta\boldsymbol{B}$ are the discrete-time matrices obtained from the $\boldsymbol{A}$ and $\boldsymbol{B}$.

Recently, Mamba has introduced selective SSMs that captures contextual information in long sequences using time-varying parameters (Gu & Dao, 2023). Its near-linear complexity makes it an efficient alternative to the quadratic complexity of the attention mechanism in Transformers.

**Problem definition.** In multivariate TS forecasting, a model uses a lookback window $\mathbf{x} = (\mathbf{x}_1, \mathbf{x}_2, \cdots, \mathbf{x}_L)$ to predict future values $\mathbf{y} = (\mathbf{x}_{L+1}, \cdots, \mathbf{x}_{L+H})$ with $\mathbf{x}_i \in \mathbb{R}^C$, representing the values at each time step. Here, $L$, $H$, and $C$ denote the size of the lookback window, the forecast horizon, and the number of channels, respectively. General forecasting models follow the framework illustrated in Figure 3, consisting of an embedding layer, an encoder for modeling CD/TD, and a prediction head. The proposed method employs Mamba as the encoder for CD, aligning with recent works (Wang et al., 2025; Liang et al., 2024).

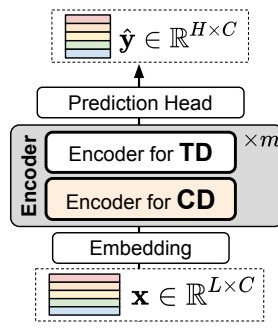

Figure 3: TS forecasting.

# 4 METHODOLOGY

In this section, we introduce FSMamba, a TS forecasting method based on a *unidirectional* Mamba designed to address the sequential order bias by 1) *regularizing Mamba* to minimize the distance between two embedding vectors generated from data with reversed channel orders and 2) *removing the 1D-conv* from the Mamba block. Furthermore, we introduce a pretraining task, **channel similarity modeling** (CSM), where the model is pretrained to preserve the similarity between channels from the data space to the latent space, aligning with the recent models focusing on capturing CD over TD.

## 4.1 ARCHITECTURE OF FSMAMBA

For the **1) embedding layer**, we use a single linear layer to tokenize the TS in a channel-wise manner (i.e., each channel as a token), following the previous works (Liu et al., 2024a; Wang et al., 2025). The resulting embeddings are passed to the encoder, where each layer consists of **2) encoder for CD** and **3) encoder for TD**. For the encoder for CD, we apply the proposed *CD-Mamba block*, which incorporates the regularization (Sec. 4.2) and the removal of 1D-conv (Sec. 4.3) to handle the bias. For the encoder for TD, we apply an MLP to the output tokens of the CD-Mamba block and use layer normalization before and after the MLP, following previous works (Liu et al., 2024a; Wang et al., 2025). Finally, for the **4) prediction head**, we employ a linear layer on the output tokens of MLP.

## 4.2 REGULARIZATION STRATEGY

To address the sequential order bias, FSMamba regularizes Mamba to minimize the distance between two embedding vectors generated with reversed channel orders. This is intuitive, as it encourages the encoder to produce similar representations regardless of the scan direction, thereby promoting robustness to channel order. The regularization term with a distance metric $d$ is defined as $L_{\text{reg}}(\mathbf{z}) = d(\mathbf{z}_1, \mathbf{z}_2)$, where $\mathbf{z}_1$ and $\mathbf{z}_2$ are the embedding vectors obtained from Mamba with its channel order reversed, as shown in Figure 4. For $d$, we use the mean squared error (MSE), with the robustness to the choice of $d$ shown in Appendix I. The regularization term is then added to the forecasting loss ($L_{\text{fcst}}$) with a contribution of $\lambda$ as:

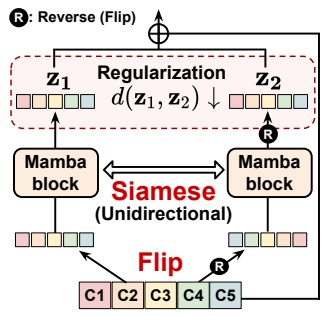

Figure 4: Proposed arch with reg.

$$L(\mathbf{x}, \mathbf{y}) = L_{\text{fcst}}(\mathbf{x}, \mathbf{y}) + \lambda \cdot \sum_{i=1}^{m} L_{\text{reg}}(\mathbf{z}^{(i)}), \tag{3}$$

where $\mathbf{z}^{(i)}$ is the embedding vector at the $i$-th layer, and $m$ is the number of encoder layers.

By regularizing unidirectional Mamba, we achieve better performance and efficiency compared to bidirectional Mamba (see Table 6 for further analysis). Additionally, we find that the regularization also benefits bidirectional Mamba, which already handles the bias through bidirectional scanning (see Table 8 for further analysis). Further analysis regarding the robustness to $\lambda$ is shown in Table 14.

## 4.3 REMOVAL OF 1D-CONVOLUTION

The original Mamba block combines the H3 block (Fu et al., 2023) with a gated MLP, where the H3 block incorporates a 1D-conv before the SSM layer to capture local information from adjacent steps of sequential data. However, since channels in TS generally do not possess any sequential order[1], we find this convolution *unnecessary for capturing CD* in such cases.

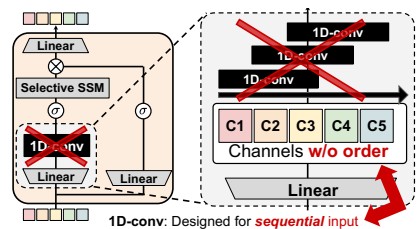

Figure 5: CD-Mamba block: w/o 1D-conv.

Accordingly, we remove the 1D-conv from the original Mamba block, resulting in the proposed *CD-Mamba block*, as illustrated in Figure 5. Using the CD-Mamba block, we obtain $\mathbf{z}_1$ and $\mathbf{z}_2$, which are two embedding vectors with reversed channel orders that are employed for regularization, as illustrated in Figure 4. These vectors are then added element-wise and combined with a residual connection from $\mathbf{z}$. Note that this removal *differs from the removal of 1D-conv in previous works* (Zeng et al., 2024; Cai et al., 2024), where the convolution was designed to capture TD instead of CD, which is unrelated to the sequential order bias. Further analysis regarding the removal of the 1D-conv can be found in Table 9.

---

[1]Metadata indicates *in advance* whether channels have an order, with *general TS datasets lacking this*.

### 4.4 CHANNEL SIMILARITY MODELING

Previous pretraining tasks for TS have primarily focused on TD, such as masked modeling (Zerveas et al., 2021) and reconstruction (Lee et al., 2024). However, we argue for the necessity of a new task that emphasizes CD over TD to align with recent trends that focus on CD (Liu et al., 2024a; Wang et al., 2025). To this end, we propose CSM, which aims to preserve the similarity between channels from the data space to the latent space.

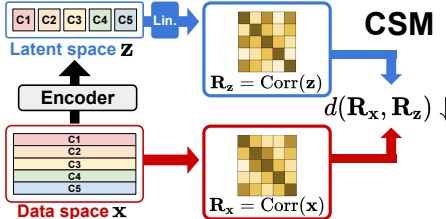

For CSM, we compute similarity matrices between the input token on the data space and the output token after the linear projection layer on the latent space, as shown in Figure 6. To preserve similarities across the two spaces, we minimize the distance between these matrices, where the loss function for CSM is defined as $L_{\text{CSM}}(\mathbf{x}) = d_{\text{CSM}}(\mathbf{R_x}, \mathbf{R_z})$, where $\mathbf{R_x}$ and $\mathbf{R_z}$ denote the correlation matrices in the data space and the latent space, respectively. Here, $d_{\text{CSM}}$ denotes the distance metric, where we employ Pearson correlation for the experiments, as it is widely used in previous works as a simple yet effective way to measure channel relation-

Figure 6: **Channel similarity modeling.** Distance between similarity matrices in the data space and the latent space is minimized.

ships (Yang et al., 2024; Zhao & Shen, 2024). We find that CSM is more effective than masked modeling and reconstruction across diverse datasets with varying numbers of channels, as shown in Table 11. Robustness to the choice of $d_{\text{CSM}}$ and the pseudocode are described in Appendix I and G.

## 5 EXPERIMENTS

### 5.1 EXPERIMENTAL SETTINGS

**Tasks and metrics.** We demonstrate the effectiveness of FSMamba on TS forecasting tasks with 13 datasets. For self-supervised learning (SSL), we follow the standard framework of pretraining and fine-tuning (FT) or linear probing (LP) on the same dataset. Additionally, we consider in- and cross-domain transfer learning settings, with the domains defined in the previous work (Dong et al., 2023). For evaluation metrics, we employ mean squared error (MSE) and mean absolute error (MAE).

**Datasets.** For the forecasting tasks, we use 13 datasets: ETT datasets (ETTh1, h2, m1,m2) (Zhou et al., 2021), PEMS datasets (PEMS03, 04, 07, 08) (Chen et al., 2001), Exchange, Weather, Traffic, ECL (Wu et al., 2021), and Solar (Lai et al., 2018). Details of the statistics are provided in Appendix A.

**Baseline methods.** We follow the baseline methods and results from S-Mamba (Wang et al., 2025). where we consider Transformer-based models, including iTransformer (Liu et al., 2024a), PatchTST (Nie et al., 2023), and Crossformer (Zhang & Yan, 2023), as well as CNN/GNN/MLP models, including TimesNet (Wu et al., 2023), CrossGNN (Huang et al., 2023), DLinear (Zeng et al., 2023), RLinear (Li et al., 2023), and TiDE (Das et al., 2023). For Mamba-based model, we use S-Mamba (Wang et al., 2025) and FACTS (Nanbo et al., 2025).

**Experimental setups.** We follow the experimental setups from iTransformer and S-Mamba. For dataset splitting, we adhere to the standard protocol of dividing all datasets into training, validation, and test sets in chronological order. Details of the setups (e.g., $L, H$) are provided in Appendix A.

### 5.2 TIME SERIES FORECASTING

Table 3 presents the results for the multivariate TS forecasting task, showing the average MSE/MAE across four horizons ($H$s) over five runs. The results demonstrate that our proposed FSMamba outperforms the SoTA Transformer-based models and S-Mamba, which uses the bidirectional Mamba, whereas our approach utilizes the unidirectional Mamba, providing greater efficiency (see Table 13 for further analysis). Additionally, due to the capacity-robustness trade-off (Han et al., 2023), CI and CD models generally benefit more from smaller and larger datasets, respectively. Nevertheless, our method outperforms both models in both settings, as shown in Figure 2.

Furthermore, comparisons with recent Mamba-based methods, including concurrent works, are presented in Table 4. The table shows that our method achieves competitive performance using unidirectional Mamba without relying on attention mechanisms (Xu et al., 2024; Xiong et al., 2025) or additional techniques such as optimal scan order search for inference (Cai et al., 2024).

| Models | | | | | | | | | | | | | | | | | | | | | | | | | | |
|---|---|---|---|---|---|---|---|---|---|---|---|---|---|---|---|---|---|---|---|---|---|---|---|---|---|---|
| | **Channel Dependent (CD)** | | | | | | | | | | | | | | | | | | **Channel Independent (CI)** | | | | | | | |
| | **Mamba** | | | | | | | **Transformer** | | | | | | **Others** | | | | **Transformer** | | **Linear/MLP** | | | | | | |
| | FSMamba (Ours) | | | | S-Mamba | | FACTS | | iTransformer | | Minusformer | | Crossformer | | TimesNet | | CrossGNN | | PatchTST | | DLinear | | RLinear | | TiDE | |
| | SSL | | SL | | (NC*2025) | | (ICLR 2025) | | (ICLR 2024) | | (arXiv 2024) | | (ICLR 2023) | | (ICLR 2023) | | (NeurIPS 2023) | | (ICLR 2023) | | (AAAI 2023) | | (arXiv 2023) | | (TMLR 2023) | |
| Metric | MSE | MAE | MSE | MAE | MSE | MAE | MSE | MAE | MSE | MAE | MSE | MAE | MSE | MAE | MSE | MAE | MSE | MAE | MSE | MAE | MSE | MAE | MSE | MAE | MSE | MAE |
| ETTh1 | **.430** | .434 | .434 | .436 | .457 | .452 | .441 | **.428** | .457 | .449 | .463 | .452 | .529 | .522 | .458 | .450 | .437 | .434 | .469 | .454 | .456 | .452 | .446 | .434 | .541 | .507 |
| ETTh2 | .376 | .403 | .378 | .405 | .383 | .408 | .376 | .398 | .384 | .407 | .394 | .409 | .942 | .684 | .414 | .427 | .393 | .418 | .387 | .407 | .559 | .515 | .374 | .398 | .611 | .550 |
| ETTm1 | .387 | .398 | .392 | .400 | .398 | .407 | .393 | .399 | .408 | .412 | .416 | .412 | .513 | .496 | .400 | .406 | .393 | .404 | .387 | .400 | .403 | .407 | .414 | .407 | .419 | .419 |
| ETTm2 | .280 | .326 | .281 | .326 | .290 | .333 | .281 | .326 | .293 | .337 | .285 | .328 | .757 | .610 | .291 | .333 | .282 | .330 | .281 | .326 | .350 | .401 | .286 | .327 | .358 | .404 |
| PEMS03 | .120 | .228 | .136 | .243 | .133 | .240 | - | - | .142 | .248 | .138 | .245 | .169 | .291 | .147 | .281 | - | - | .180 | .248 | .278 | .375 | .495 | .472 | .326 | .419 |
| PEMS04 | .099 | .203 | .103 | .211 | .103 | .211 | - | - | .121 | .232 | .171 | .270 | .209 | .314 | .129 | .241 | - | - | .195 | .307 | .295 | .388 | .526 | .491 | .353 | .437 |
| PEMS07 | .089 | .187 | .090 | .191 | .090 | .191 | - | - | .102 | .205 | .125 | .224 | .235 | .315 | .124 | .225 | - | - | .211 | .303 | .329 | .395 | .504 | .478 | .380 | .440 |
| PEMS08 | .133 | .225 | .148 | .236 | .157 | .242 | - | - | .254 | .306 | .302 | .358 | .268 | .307 | .193 | .271 | - | - | .280 | .321 | .379 | .416 | .529 | .487 | .441 | .464 |
| Exchange | .358 | .403 | .361 | .404 | .364 | .407 | .355 | .398 | .368 | .409 | .508 | .488 | .940 | .707 | .416 | .443 | .345 | .395 | .367 | .404 | .354 | .414 | .378 | .417 | .370 | .413 |
| Weather | .244 | .271 | .246 | .274 | .252 | .277 | .250 | .277 | .260 | .281 | .260 | .281 | .259 | .315 | .259 | .287 | .247 | .289 | .259 | .281 | .265 | .317 | .272 | .291 | .271 | .320 |
| Solar | .230 | .259 | .233 | .259 | .244 | .275 | .256 | .274 | .234 | .261 | .230 | .253 | .641 | .639 | .301 | .319 | - | - | .270 | .307 | .330 | .401 | .369 | .356 | .347 | .417 |
| ECL | .163 | .259 | .163 | .256 | .174 | .269 | .168 | .264 | .179 | .270 | .171 | .262 | .244 | .334 | .192 | .295 | .201 | .300 | .205 | .290 | .212 | .300 | .219 | .298 | .251 | .344 |
| Traffic | .394 | .269 | .399 | .269 | .417 | .277 | .470 | .298 | .428 | .282 | .413 | .272 | .550 | .304 | .620 | .336 | .583 | .323 | .481 | .304 | .625 | .383 | .626 | .378 | .760 | .473 |
| Average | .254 | .297 | .259 | .301 | .266 | .307 | - | - | .278 | .315 | .288 | .319 | .481 | .448 | .306 | .338 | .303 | .329 | .372 | .397 | .418 | .403 | .418 | .431 | | |
| 1st Count | 29 | 25 | 9 | 11 | 4 | 4 | 6 | 11 | 0 | 0 | 2 | 2 | 1 | 0 | 0 | 0 | 3 | 3 | 5 | 2 | 2 | 0 | 3 | 3 | 0 | 0 |
| 2nd Count | 18 | 16 | 24 | 22 | 4 | 5 | 4 | 7 | 0 | 0 | 2 | 1 | 0 | 0 | 0 | 0 | 2 | 2 | 0 | 4 | 0 | 0 | 0 | 6 | 0 | 0 |
| 1st or 2nd | 47 | 41 | 33 | 33 | 8 | 9 | 10 | 18 | 4 | 2 | 4 | 3 | 1 | 0 | 0 | 0 | 5 | 5 | 5 | 6 | 4 | 0 | 3 | 9 | 0 | 0 |

Table 3: **Results of TS forecasting.** We compare our method with the SoTA methods with $L = 96$. The best results are in **bold** and the second best are underlined. For Mamba-based methods, we include only recently peer-reviewed models, with comparisons with other methods presented in Table 4. (NC*: *Neurocomputing*)

| Model | **Mamba for TD** ($\mathcal{O}(L)$) | | | | | | **Mamba for TD & CD** ($\mathcal{O}(L \cdot C)$ or $\mathcal{O}(L+C)$) | | | | | **Mamba for CD** ($\mathcal{O}(C)$) | | | | |
|---|---|---|---|---|---|---|---|---|---|---|---|---|---|---|---|---|
| | CMamba | Heracles | SiMBA | msMamba | SST(1) | FACTS | MambaMixer | SAMBA | MTS-UNMixer | MambaTS(2) | TimeMachine(3) | FMamba | Att.Mamba | Bi-Mamba+ | **S-Mamba** | **FSMamba** |
| Venue | arXiv'24 | arXiv'24 | arXiv'24 | arXiv'25 | arXiv'24 | ICLR'25 | arXiv'24 | arXiv'24 | arXiv'24 | arXiv'24 | ECAI'24 | arXiv'24 | arXiv'24 | arXiv'25 | NC'25 | Ours |
| Code | | ✗ | | | O | O | | ✗ | | O | O | ✗ | | ✗ | ■ | O |
| How to handle the sequential order bias? ✗ Does not handle, [■] Bidirectional, [▲] Additional training, [★] Regularization | | | | | | | ■ | ■ | ■ | ▲ | ✗ | ✗ | ■ | ■ | ■ | ★ |
| ETTh1 | .433 | .435 | .442 | .445 | .439 | .441 | .404 | .443 | .423 | - | .433 | - | - | .437 | .457 | .430 |
| ETTh2 | .368 | .364 | .362 | .373 | .363 | .376 | .334 | .363 | .345 | .357 | .347 | - | - | .372 | .383 | .376 |
| ETTm1 | .376 | .398 | .383 | .394 | .362 | .393 | .361 | .378 | .389 | - | .383 | - | - | .378 | .398 | .387 |
| ETTm2 | .273 | .283 | .282 | .284 | .272 | .281 | .268 | .276 | .274 | .264 | .272 | - | - | .281 | .290 | .280 |
| Weather | .237 | .276 | .256 | .249 | .228 | .250 | .240 | .249 | .239 | .225 | .244 | .247 | .247 | .244 | .252 | .244 |
| Solar | - | - | - | .231 | - | .256 | - | .229 | - | - | .250 | - | .231 | .227 | .244 | .230 |
| ECL | .169 | .173 | .185 | .165 | .170 | .264 | .172 | .172 | .176 | .156 | .170 | - | .167 | .166 | .174 | .170 |
| Traffic | .444 | - | .469 | .406 | .400 | .470 | .420 | .422 | .466 | .373 | .429 | - | - | .404 | .417 | .394 |
| Avg. | - | - | - | .381 | - | .341 | - | - | - | .318 | .316 | - | - | .314 | .329 | **.313** |

**(1)** While our method uses only Mamba, SST combines *both Transformer and Mamba*.

**(2)** MambaTS requires an *additional procedure to learn the optimal scan order for inference* during training.

**(3)** For fair comparison, we apply the *CD arch for all datasets*, whereas the original paper uses CI or CD depending on the dataset.

Table 4: **Comparison with Mamba-based methods.** We compare our method with recent Mamba-based methods including 15 concurrent works. Note that we exclude the PEMS datasets (Chen et al., 2001), as many algorithms do not use them in their experiments.

| | | | LP | | | FT | | |
|---|---|---|---|---|---|---|---|---|
| | Source | Target | Uni. | Bi. | Imp. | Uni. | Bi. | Imp. |
| In-domain | ETTh2 | ETTh1 | .452 | .450 | -0.4% | .430 | .464 | 7.3% |
| | ETTm2 | ETTm1 | .396 | .398 | 0.5% | .388 | .400 | 3.0% |
| Cross-domain | ETTm2 | ETTh1 | .449 | .450 | 0.2% | .435 | .455 | 4.5% |
| | ETTh2 | ETTm1 | .396 | .401 | 1.2% | .388 | .402 | 3.5% |
| | ETTm1 | ETTh1 | .450 | .450 | 0.0% | .432 | .468 | 7.7% |
| | ETTh1 | ETTm1 | .400 | .403 | 0.7% | .389 | .399 | 2.5% |
| | Weather | ETTh1 | .449 | .546 | 17.8% | .432 | .552 | 21.7% |
| | Weather | ETTm1 | .395 | .460 | 14.1% | .388 | .501 | 22.6% |

Table 5: Transfer learning with CSM.

| Improve (↑) | Forecast horizon ($H$) | | | | Average | | # Parameters |
|---|---|---|---|---|---|---|---|
| | 96 | 192 | 336 | 720 | MSE | Imp. | ($L, H = 96$) |
| S-Mamba | .385 (-) | .445 (-) | .491 (-) | .506 (-) | .457 (-) | - | 9.29M |
| + Reg. | .381 (↑) | .433 (↑) | .476 (↑) | .488 (↑) | .444 (↑) | 2.8% | 9.29M |
| + Bi → Uni | .377 (↑) | .427 (↑) | .472 (↑) | .482 (↑) | .440 (↑) | 0.9% | 5.81M |
| - 1D-conv | .377 (-) | .426 (↑) | .468 (↑) | .464 (↑) | .434 (↑) | 0.7% | 5.80M |
| + CSM | **.372** (↑) | **.424** (↑) | **.466** (↑) | **.459** (↑) | **.430** (↑) | 0.9% | 5.80M |

Table 6: Ablation study of **Reg.**, **Model** and **Pretraining**.

## 5.3 TRANSFER LEARNING

To assess the transferability of FSMamba, we conduct experiments using CSM in both in- and cross-domain transfer settings following SimMTM (Dong et al., 2023), where source and target datasets share the same frequency in the in-domain setting, while not in the cross-domain setting. Table 5 presents the average MSE across four $H$s, demonstrating that FSMamba consistently outperforms S-Mamba, especially in cross-domain settings where the source and target datasets differ significantly.

## 6 ABLATION STUDIES

**Summary of results.** To demonstrate the effectiveness of our method, we conduct an ablation study with ETTh1 to evaluate the impact of the following components: 1) adding the regularization term, 2) using the unidirectional Mamba instead of the bidirectional Mamba, 3) removing the 1D-conv, and 4) pretraining with CSM. Table 6 presents the results, indicating that using all proposed components results in the best performance with 37.6% fewer model parameters compared to S-Mamba.

**Architecture for TD & CD.** Following the recent studies (Liu et al., 2024a; Wang et al., 2025) that suggest employing simple models to capture TD in TS, we utilize an MLP for this purpose. To examine the impact of different design choices of architecture for capturing TD, we consider two alternatives: 1) without employing any encoder for TD, and 2) using Mamba, following the previous work (Wang et al., 2025). Additionally, we compare methods with various CD architectures, keeping the MLP fixed for the TD architecture. Table 7 shows that our method achieves the best performance with an MLP, which aligns with previous works using simple models for TS and complex models for CD, and *justifies the use of Mamba for CD instead of TD*, even with the sequential order-bias issue.

| Architecture for | | ETT | | | | PEMS | | | | Exchange | Weather | Solar | ECL | Traffic | Avg. |
|---|---|---|---|---|---|---|---|---|---|---|---|---|---|---|---|
| TD | CD | h1 | h2 | m1 | m2 | 03 | 04 | 07 | 08 | | | | | | |
| - | | .440 | .386 | .369 | .286 | .137 | .105 | .097 | .154 | **.361** | .248 | .236 | .166 | .416 | .262 |
| Mamba | Mamba | .440 | .382 | .371 | .284 | .140 | .106 | .096 | .155 | .362 | .254 | .239 | .165 | .410 | .262 |
| MLP | | **.434** | **.378** | **.392** | **.281** | **.136** | **.103** | **.090** | **.148** | **.361** | **.246** | **.233** | **.163** | **.399** | **.257** |
| | -(1) | .446 | **.374** | .403 | .286 | .278 | .295 | .329 | .379 | .354 | .265 | .330 | .212 | .625 | .352 |
| MLP | MLP(2) | .462 | .403 | .401 | .287 | **.129** | .115 | .115 | .186 | .365 | .260 | .255 | .211 | .498 | .284 |
| | Trans.(3) | .457 | .384 | .408 | .293 | .142 | .121 | .102 | .254 | .368 | .260 | .234 | .178 | .428 | .278 |
| | Mamba | **.434** | .378 | **.392** | **.281** | .136 | **.103** | **.090** | **.148** | **.361** | **.246** | **.233** | **.163** | **.399** | **.257** |

Table 7: **Architecture for TD & CD.** Consistent with prior works (Liu et al., 2024a; Wang et al., 2025), a complex model may not be necessary for capturing TD. We report the lower MSE between DLinear and RLinear for (1), the MSE of TSMixer (Chen et al., 2023) for (2), and the MSE of iTransformer for (3).

| Method | Mamba | | ETT | | | | PEMS | | | | Exchange | Weather | Solar | ECL | Traffic | # Best (/52) | Avg. | Imp.(%) |
|---|---|---|---|---|---|---|---|---|---|---|---|---|---|---|---|---|---|---|
| | # | Reg. | h1 | h2 | m1 | m2 | 03 | 04 | 07 | 08 | | | | | | | | |
| S-Mamba | Bi | ✗ | .457 | .383 | .398 | .290 | .133 | .103 | .090 | .157 | .364 | .252 | .244 | .174 | .417 | 0 | .266 | - |
| - | | ✓ | **.444** | **.377** | **.386** | **.277** | **.131** | **.096** | **.084** | **.135** | **.359** | **.249** | **.232** | **.168** | **.404** | 52 | .259 | +2.6% |
| - | Uni | ✗ | .438 | .380 | .396 | .283 | .146 | .104 | .092 | .283 | .364 | .255 | .237 | .165 | .411 | 0 | .274 | - |
| FSMamba | | ✓ | **.434** | **.378** | **.392** | **.281** | **.136** | **.103** | **.090** | **.148** | **.361** | **.246** | **.233** | **.163** | **.399** | 52 | .259 | +5.5% |

Table 8: **Proposal 1: Effect of regularization.** Regularization enhances *both unidirectional and bidirectional* Mamba. For unidirectional Mamba, we remove 1D-conv for both w/ and w/o Reg. for fair comparison.

| Method | Mamba | | ETT | | | | PEMS | | | | Exchange | Weather | Solar | ECL | Traffic | # Best (/52) | Avg. | Imp.(%) |
|---|---|---|---|---|---|---|---|---|---|---|---|---|---|---|---|---|---|---|
| | # | 1D-conv | h1 | h2 | m1 | m2 | 03 | 04 | 07 | 08 | | | | | | | | |
| S-Mamba | Bi | ✓ | .457 | .383 | .398 | .290 | .133 | .103 | **.090** | .157 | .364 | .252 | .244 | .174 | .417 | 10 | .266 | - |
| - | | ✗ | **.452** | **.382** | **.394** | **.286** | **.131** | **.096** | .092 | **.155** | **.361** | **.251** | **.242** | **.170** | **.411** | 42 | .263 | +1.1% |
| - | Uni | ✓ | .440 | .380 | .398 | **.281** | **.136** | **.099** | **.091** | **.143** | **.361** | .253 | .234 | .168 | .402 | 14 | .261 | - |
| FSMamba | | ✗ | **.434** | **.378** | **.392** | **.281** | **.136** | .103 | **.091** | .148 | **.361** | **.246** | **.233** | **.163** | **.399** | 38 | .259 | +0.8% |

Table 9: **Proposal 2: Effect of removal of 1D-conv.** Using 1D-conv generally degrades performance as *channels lack a sequential order in general*. PEMS is an exception where channels follow a geographical order which can be known in advance via metadata, allowing 1D-conv to be retained, though it does not always improve performance. For unidirectional Mamba, we apply regularization for both cases for fair comparison.

**Effect of regularization.** To validate the effect of the regularization strategy, we apply it to both the unidirectional and the bidirectional Mamba. The results are shown in Table 8, which presents the average MSE across four $H$s. These results indicate that it not only improves the performance of the unidirectional Mamba, but also benefits the bidirectional Mamba which already handles the bias, making it complementary to the bidirectional scanning. Furthermore, by considering only a given order and its reverse, the model can efficiently induce interactions across all tokens, further highlighting the advantage of the regularization strategy, which is further discussed around Figure 12.

**Effect of removal of 1D-conv.** To examine the role of the 1D-conv in Mamba for capturing CD, we remove it from both unidirectional and bidirectional Mamba, with the results of the average MSE across four $H$s shown in Table 9. The results indicate that, for general datasets where channels lack a sequential order, *removing the 1D-conv does not significantly affect performance*. Rather, it shows benefits on most datasets, with the exception of PEMS datasets (Liu et al., 2022), where channels follow a geographical order that can be known in advance.

# 7 ANALYSIS

In this section, we analyze the effectiveness of our method across various aspects: **[a–b]** Handling the sequential order bias, **[c]** Efficiency analysis, 3) **[d–f]** Impact of CSM, and **[g–j]** Others.

**[a] Bias by dataset.** The degree of a sequential order bias may vary depending on the characteristics of the datasets. We consider two factors affecting this degree: 1) the *correlation between channels* and 2) the *number of channels* ($C$). To evaluate the relationships between these factors and the degree of bias, we quantify the degree of bias for each dataset as the difference in performance (average MSE across four $H$s) when the channel order is reversed, using FSMamba without regularization.

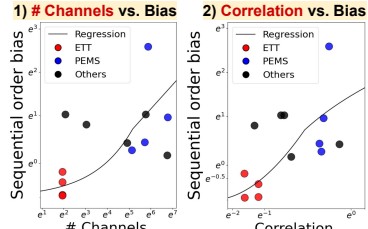

Figure 7: **[a]** Varying bias by datasets.

Figure 7 shows the results with two plots, where the horizontal axes are $C$ and correlation between the channels (i.e., average of the off-diagonal elements in the correlation matrix[2]) between the channels), and the vertical axes represent the degree of a bias, with all axes shown on a log scale. The results show that the bias increases 1) as the channels become

---

[2]We use its absolute value, as high correlation does not always indicate a strong relationship.

| Dataset | FSMamba (SSL) | | | | | S-Mamba |
|---|---|---|---|---|---|---|
| | Cosine | $\ell_1$ | $\ell_2$ | Corr-G | Corr-L | |
| ETTh1 | .432 | .431 | **.430** | .434 | **.430** | .457 |
| ETTh2 | **.376** | .377 | .376 | .378 | **.376** | .383 |
| ETTm1 | .388 | .387 | .388 | .390 | .387 | .398 |
| ETTm2 | .281 | .281 | **.280** | .281 | .280 | .290 |
| PEMS03 | **.119** | .122 | .121 | .123 | .120 | .133 |
| PEMS04 | .103 | .100 | .099 | .101 | .099 | **.096** |
| PEMS07 | .090 | **.088** | **.088** | .089 | .089 | .090 |
| PEMS08 | .133 | .135 | **.132** | .139 | .133 | .157 |
| Exchange | .360 | .360 | .358 | .360 | .358 | .364 |
| Weather | **.244** | **.244** | **.244** | .246 | **.244** | .252 |
| Solar | **.229** | .230 | .231 | .230 | .230 | .244 |
| ECL | **.163** | **.163** | **.163** | **.163** | **.163** | .174 |
| Traffic | .397 | .396 | .395 | .396 | **.394** | .417 |
| Average | .255 | .255 | **.254** | .256 | **.254** | .266 |

Table 10: **[d]** Robustness to similarity metrics for CSM.

| Dataset | FSMamba | | | | S-Mamba | | | |
|---|---|---|---|---|---|---|---|---|
| | SL | SSL | | | SL | SSL | | |
| | | Rec. | MM | CSM | | Rec. | MM | CSM |
| ETTh1 | .434 | .432 | .433 | **.430** | .457 | .448 | .457 | .457 |
| ETTh2 | .378 | .377 | .378 | **.376** | .383 | .381 | .383 | **.380** |
| ETTm1 | .392 | **.389** | .391 | **.389** | .398 | .400 | .397 | **.396** |
| ETTm2 | .281 | **.279** | .281 | .280 | .290 | **.283** | .288 | .286 |
| PEMS03 | .136 | .125 | .121 | **.120** | .133 | .120 | .130 | .119 |
| PEMS04 | .103 | .102 | **.095** | .099 | .096 | **.092** | .103 | .093 |
| PEMS07 | .090 | .090 | .089 | **.089** | .090 | **.086** | .089 | **.085** |
| PEMS08 | .148 | **.133** | .140 | **.133** | .157 | **.136** | .157 | .138 |
| Exchange | .361 | **.357** | .360 | .358 | .364 | .363 | .378 | **.361** |
| Weather | .246 | .246 | .247 | **.244** | .252 | .249 | .251 | .250 |
| Solar | .233 | .231 | .231 | **.230** | .244 | .230 | .239 | .233 |
| ECL | **.163** | **.163** | **.163** | **.163** | .174 | .175 | .174 | **.170** |
| Traffic | .399 | .397 | .395 | **.394** | .417 | .450 | .415 | **.414** |
| Average | .259 | .256 | .256 | **.254** | .266 | .263 | .266 | **.260** |

Table 11: **[e]** Comparison of SSL pretraining tasks.

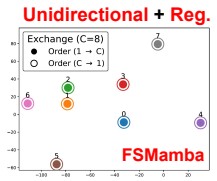 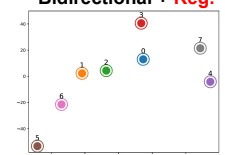 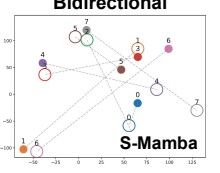

Figure 8: **[b]** t-SNE of channel representations w/ and w/o reg.

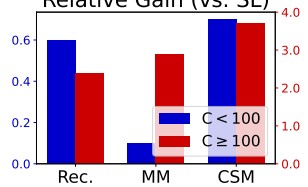

Figure 9: **[e]** SSL comparison.

more correlated and 2) as $C$ increases. For example, ETT datasets ($C = 7$) of low correlation show low bias, whereas PEMS datasets ($C > 100$) of high correlation exhibit high bias.

Furthermore, TimeMachine (Ahamed & Cheng, 2024), which *does not* handle the bias, performs worse on *datasets with large $C$s* (e.g., Solar, ECL) compared to small $C$s (e.g., ETTs), as shown in Table 4. This highlights the importance of handling the bias in datasets especially with large $C$s.

**[b] Robustness to channel order.** To demonstrate that our method effectively addresses a sequential order bias, we conduct two analyses to show its robustness to the channel order. First, we evaluate the performance variations with five random permutations of channel order using ETTh1, where our method achieves a smaller standard deviation compared to S-Mamba, as shown in Table 12. Second, we visualize the output tokens of the encoder (i.e., embedding vectors of each channel) using t-SNE (Van der Maaten & Hinton, 2008) with Exchange. Figure 8 shows that the tokens from the two views with reversed orders are consistent with regularization, while inconsistent without it.

| $H$ | Uni. + Reg. | Bi. |
|---|---|---|
| 96 | $.377_{\pm.0002}$ | $.386_{\pm.0010}$ |
| 192 | $.426_{\pm.0002}$ | $.440_{\pm.0033}$ |
| 336 | $.468_{\pm.0002}$ | $.484_{\pm.0046}$ |
| 720 | $.464_{\pm.0003}$ | $.502_{\pm.0057}$ |

Table 12: **[b]** Channel order robustness.

**[c] Efficiency analysis.** To demonstrate the efficiency of FSMamba, we compare it with iTransformer and S-Mamba in terms of 1) the number of parameters, 2) memory usage, and 3) computational time with the Traffic dataset. Table 13 shows the results, indicating that FSMamba outperforms these methods in all three aspects, particularly reducing the number of parameters by up to 38.1% compared to S-Mamba. Note that the training time is measured per epoch, while the inference time is measured per data instance.

| Dataset: Traffic $(L, H = 96)$ | (a) iTrans. | (b) S-Mamba | (c) FSMamba | (b) → (c) Imp. |
|---|---|---|---|---|
| **# Params.** | | | | |
| Embedding | 0.05M | 0.05M | 0.05M | - |
| Encoder for CD | 4.20M | 6.97M | **3.48M** | **50.1%** |
| Encoder for TD | 2.11M | 2.11M | 2.11M | - |
| Pred. head | 0.05M | 0.05M | 0.05M | - |
| Total | 6.52M | 9.29M | **5.80M** | **38.1%** |
| **Memory** | | | | |
| Complexity | $\mathcal{O}(C^2)$ | $\mathcal{O}(C)$ | $\mathcal{O}(C)$ | - |
| GPU mem. (GB) | 1.36 | 0.33 | **0.32** | **4.2%** |
| **Computational time** | | | | |
| Train (sec.) | 115.5 | 108.3 | **102.1** | **5.7%** |
| Inference (ms) | 14.6 | 9.9 | **8.7** | **11.3%** |
| Avg. MSE ($4H$s) | 0.428 | 0.417 | **0.402** | **3.6%** |

Table 13: **[c]** Efficiency analysis.

**[d] Robustness to CSM metrics.** To demonstrate the robustness of the similarity metric for CSM, we evaluate using different metrics including cosine similarity and (negative) $\ell_1$ and $\ell_2$ distances. Table 10 presents the TS forecasting results with average MSE across four $H$s, indicating that performance is robust to the metric. Furthermore, for correlation metric, we consider two candidates: *local* correlation (Corr-L) (i.e., the correlation between the channels of the *input TS*) and *global* correlation (Corr-G) (i.e., the correlation between the channels of the *entire dataset*). The table shows that using the local correlation yields better performance, although both approaches still outperform the competitive baseline (Gu & Dao, 2023).

**[e] Effect of CSM.** To demonstrate the effect of CSM, we compare it with two other pretraining tasks: masked modeling (MM) (Zerveas et al., 2021) with a masking ratio of 50% and reconstruction (Rec.) (Lee et al., 2024). Table 11 presents the results showing that CSM outperforms the other tasks

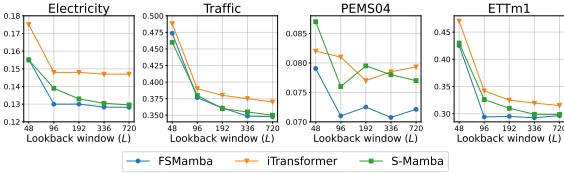

Figure 10: **[f]** Viz of $\mathbf{R}$s.

Figure 11: **[f]** $d_{\mathrm{CSM}}(\mathbf{R_x}, \mathbf{R_z})$.

Figure 12: **[g]** Reg loss.

Figure 13: **[i]** Missingness.

Figure 14: **[h] Various $L$s.** MSE with $L \in \{48, 96, 192, 336, 720\}$ and $H = 12$ for PEMS04 and $H = 96$ for others.

Table 14: **[j]** Robustness to $\lambda$ for reg.

| Dataset | FSMamba (SL) | | | | | | | S-Mamba |
|---|---|---|---|---|---|---|---|---|
| | w/o reg. | w/ reg. | | | | | | |
| | 0 | 0.0001 | 0.001 | 0.01 | 0.1 | 0.2 | 0.5 | |
| ETTh1 | .455 | .437 | **.434** | **.434** | **.434** | **.434** | **.434** | .457 |
| ETTh2 | .383 | .380 | **.378** | **.378** | **.378** | **.378** | **.378** | .383 |
| ETTm1 | .403 | .394 | **.392** | **.392** | **.392** | **.392** | .393 | .398 |
| ETTm2 | .289 | .283 | **.281** | **.281** | **.281** | **.281** | .282 | .290 |
| Avg. | .383 | .374 | **.371** | **.371** | **.371** | **.371** | .372 | .382 |

on both S-Mamba and FSMamba. Furthermore, as CSM is designed to effectively capture CD in datasets, we compare the performance gain from three tasks based on the *number of channels*, with six datasets of $C < 100$ and seven datasets of $C \geq 100$. Figure 9 shows the average performance gain from FT with three tasks compared to SL, indicating that reconstruction excels in performance with fewer *channels* and MM excels with more channels, while CSM outperforms in both cases.

**[f] Correlation in the data space and the latent space.** To demonstrate that CSM effectively preserves the relationships between channels from the data space to the latent space, we visualize the correlation matrices in both spaces with FSMamba pretrained with CSM. Figure 10 shows the results on the Weather dataset, which indicate that the relationships are effectively preserved with CSM. Additionally, we compare the distances between the matrices in both spaces, comparing FSMamba without pretraining to the one pretrained with CSM. The results, illustrated in Figure 11, show that the model pretrained with CSM exhibits a smaller difference between the matrices.

**[g] Channel order: Fixed vs. Random.** For the regularization strategy, the distance between two vectors derived from reversed channel orders is minimized, with the orders fixed across iterations. To assess the impact of fixing or permuting the order randomly across iterations, we explore four cases based on whether the channel orders of two vectors are fixed or randomly permuted in each iteration. As shown in the regularization loss curves for PEMS08 in Figure 12, maintaining a fixed channel order for each view, along with the reverse order, leads to stable training (blue line), while permuting the order results in instability (other lines). Further analysis is discussed in Appendix H.

**[h] Various sizes of lookback window ($L$).** Following the previous works (Wang et al., 2025), we conduct an experiment to evaluate the performance by the size of the lookback window ($L$) with four datasets. The results, shown in Figure 14, indicate that the performance remains robust to $L$ for some datasets and even improves with larger $L$ for others.

**[i] Robustness to missing values.** Real-world TS datasets often exhibit non-stationarity (Han et al., 2023), including scenarios with missing values. To assess the robustness of our method in these scenarios, we conduct experiments in scenarios where 25%, 50%, and 75% of values are randomly missing and interpolated using adjacent values. Figure 13 shows the average MSE across four $H$s, indicating that our method remains robust even with significant amounts of missing data and that our method trained with missing values outperforms S-Mamba trained without missingness.

**[j] Robustness to $\lambda$ for regularization.** Table 14 shows the average MSE across four different horizons on ETT datasets, using various values of $\lambda$ controlling the contribution of the regularization. The results highlight the regularization's effectiveness and its stability even with a small $\lambda$ value. This is consistent with the rapid convergence of the regularization loss early in training, as shown in Figure 12, enabling the regularization term to align the two vectors effectively with small $\lambda$.

## 8 CONCLUSION

In this work, we introduce FSMamba, a TS forecasting method that addresses the sequential order bias by incorporating a regularization strategy into unidirectional Mamba. Additionally, we propose a novel pretraining task, CSM, to improve the model's ability to capture CD. Our results demonstrate that the proposed method is robust to variations in channel order, leading to superior performance and greater efficiency. We hope that our work motivates further research on Mamba in domains where a sequential order is not inherent (e.g., tabular data).

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

APPENDIX

## A  DATASET STATISTICS AND EXPERIMENTAL SETUPS

**Dataset statistics.** We assess the performance of FSMamba across 13 datasets, with the dataset statistics detailed in Table A.1, where $C$ and $T$ denote the number of channels and timesteps, respectively.

**Experimental setups.** We follow the same data processing steps and train-validation-test split protocol as used in S-Mamba (Wang et al., 2025), maintaining a chronological order in the separation of training, validation, and test sets, using a 6:2:2 ratio for the Solar-Energy, ETT, and PEMS datasets, and a 7:1:2 ratio for the other datasets. The results are shown in Table A.1, where $N$,$L$, and $H$ represent the dataset size, the size of the lookback window, and the size of the forecast horizon, respectively. For all datasets and all models, $L$ is uniformly set to 96. We do not tune any hyperparameters and adhere to those used in S-Mamba, except for $\lambda$, which is related to the proposed regularization, and is tuned using a grid search over $[0.001, 0.01, 0.1]$.

| Dataset | Statistics | | Experimental Setups | | |
|---|---|---|---|---|---|
| | $C$ | $T$ | $(N_{\text{train}}, N_{\text{val}}, N_{\text{test}})$ | $L$ | $H$ |
| ETTh1 (Zhou et al., 2021) | | 17420 | (8545, 2881, 2881) | | |
| ETTh2 (Zhou et al., 2021) | 7 | 17420 | (8545, 2881, 2881) | | |
| ETTm1 (Zhou et al., 2021) | | 69680 | (34465, 11521, 11521) | | |
| ETTm2 (Zhou et al., 2021) | | 69680 | (34465, 11521, 11521) | | {96, 192, 336, 720} |
| Exchange (Wu et al., 2021) | 8 | 7588 | (5120, 665, 1422) | | |
| Weather (Wu et al., 2021) | 21 | 52696 | (36792, 5271, 10540) | 96 | |
| ECL (Wu et al., 2021) | 321 | 26304 | (18317, 2633, 5261) | | |
| Traffic (Wu et al., 2021) | 862 | 17544 | (12185, 1757, 3509) | | |
| Solar-Energy (Lai et al., 2018) | 137 | 52560 | (36601, 5161, 10417) | | |
| PEMS03 (Liu et al., 2022) | 358 | 26209 | (15617, 5135, 5135) | | |
| PEMS04 (Liu et al., 2022) | 307 | 15992 | (10172, 3375, 3375) | | {12, 24, 48, 96} |
| PEMS07 (Liu et al., 2022) | 883 | 28224 | (16911, 5622, 5622) | | |
| PEMS08 (Liu et al., 2022) | 170 | 17856 | (10690, 3548, 3548) | | |

Table A.1: Datasets for TS forecasting.

## B  BASELINE METHODS

- S-Mamba (Wang et al., 2025): S-Mamba utilizes the bidirectional Mamba to capture channel dependencies in TS by scanning the channels from both directions.

- PatchTST (Nie et al., 2023): PatchTST segments TS into patches and feeds them into a Transformer in a channel independent manner.

- iTransformer (Liu et al., 2024a): iTransformer reverses the conventional role of the Transformer in the TS domain by treating each channel rather than patches as a token, thereby emphasizing channel dependencies over temporal dependencies.

- Crossformer (Zhang & Yan, 2023): Crossformer employs a cross-attention mechanism to capture both temporal and channel dependencies in TS.

- TimesNet (Wu et al., 2023): TimesNet captures both intraperiod and interperiod variations in 2D space using a parameter-efficient inception block.

- RLinear (Li et al., 2023): RLinear is a simple linear model that integrates reversible normalization and channel independence.

- DLinear (Zeng et al., 2023): DLinear is a simple linear model with channel independent architecture, that employs TS decomposition.

## C  S-Mamba vs. FSMamba

Figure C.1 visualizes the comparison between S-Mamba (Wang et al., 2025), which employs the bidirectional Mamba to capture CD, and our method, FSMamba, which uses a single unidirectional Mamba with regularization to capture CD.

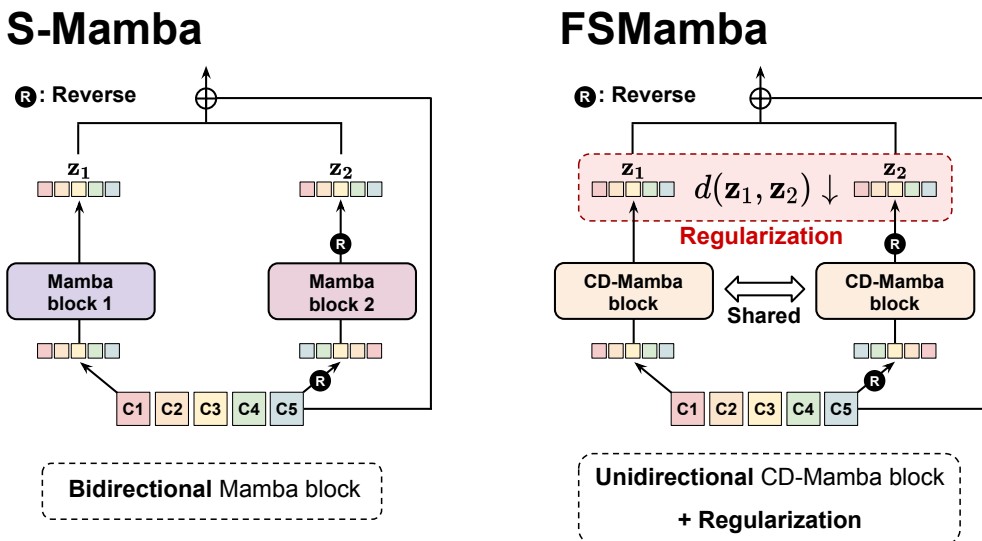

Figure C.1: Comparison of S-Mamba and FSMamba.

## D  Pseudocode of FSMamba

---

**Algorithm 1** Procedure of FSMamba

---

**Input**: $\mathbf{X} = [\mathbf{X}_1, \ldots, \mathbf{X}_L] : (B, L, C)$
**Output**: $\hat{\mathbf{Y}} = [\hat{\mathbf{X}}_{L+1}, \ldots, \hat{\mathbf{X}}_{L+H}] : (B, H, C)$
 1: $\mathbf{Z} : (B, C, D) \leftarrow \text{Linear}(\mathbf{X}^\top)$
 2: **for** $m$ in layers **do**
 3:    $\mathbf{Z}_1 : (B, C, D) \leftarrow \text{CD-Mamba}(\mathbf{Z})$
 4:    $\mathbf{Z}_2 : (B, C, D) \leftarrow \text{CD-Mamba}(\mathbf{Z}^\star)^\star$ where $\mathbf{Z}^\star = \mathbf{Z}[:, :: -1, :]$
 5:    $\mathbf{Z} : (B, C, D) \leftarrow (\mathbf{Z}_1 + \mathbf{Z}_2) + \mathbf{Z}$
 6:    $\mathbf{Z} : (B, C, D) \leftarrow \text{LN}(\text{MLP}(\text{LN}(\mathbf{Z})))$
 7: **end for**
 8: $\hat{\mathbf{Y}} : (B, H, C) \leftarrow \text{Linear}(\mathbf{Z})^\top$

---

# E    REMOVAL OF 1D-CONVOLUTION

The original Mamba block (Gu & Dao, 2023) integrates the H3 block (Fu et al., 2023) with a gated MLP, where the H3 block uses a 1D-conv before the SSM layer to capture local information within nearby tokens, as illustrated in Figure E.1. However, since channels in TS do not have an inherent sequential order, we eliminate the 1D-conv from the Mamba block, resulting in the proposed CD-Mamba block. Figure E.2 shows the overall architecture of the proposed CD-Mamba block, where the 1D-conv before the selective SSM is removed from the original Mamba block (Gu & Dao, 2023).

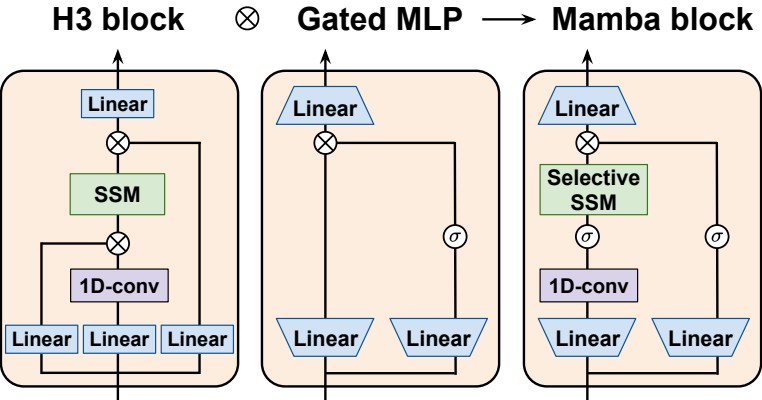

Figure E.1: **Architecture of the original Mamba block.** The original Mamba block contains 1D-conv before the SSM layer to capture local information within nearby tokens.1

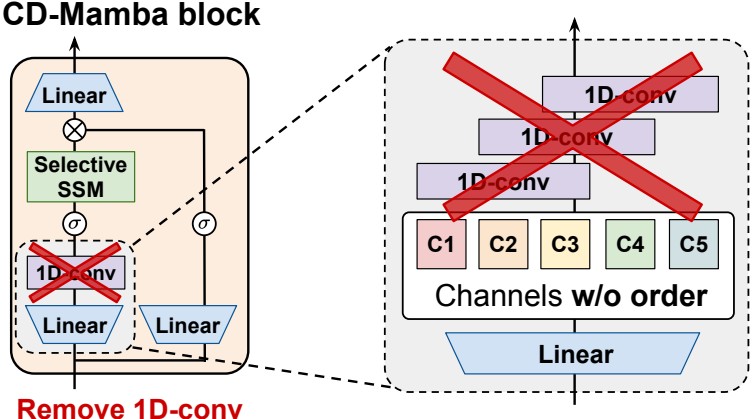

Figure E.2: **Architecture of the CD-Mamba block.** 1D-conv before the selective SSM is removed from the original Mamba block, as the channels do not have a sequential order in general.

# F FULL RESULTS OF TIME SERIES FORECASTING

Table F.1 shows the full results of TS forecasting tasks across four different horizons, highlighting the effectiveness of our method.

| Dataset | Horizon | FSMamba FT MSE | FT MAE | FSMamba SL MSE | SL MAE | S-Mamba MSE | S-Mamba MAE | FACTS MSE | FACTS MAE | Minusformer MSE | Minusformer MAE | iTransformer MSE | iTransformer MAE | RLinear MSE | RLinear MAE | PatchTST MSE | PatchTST MAE | CrossGNN MSE | CrossGNN MAE | Crossformer MSE | Crossformer MAE | TiDE MSE | TiDE MAE | TimesNet MSE | TimesNet MAE | DLinear MSE | DLinear MAE |
|---|---|---|---|---|---|---|---|---|---|---|---|---|---|---|---|---|---|---|---|---|---|---|---|---|---|---|---|
| ETTh1 | 96 | .372 | .397 | .377 | .399 | .385 | .404 | .382 | .390 | .387 | .404 | .387 | .405 | .386 | .395 | .414 | .419 | .382 | .398 | .423 | .448 | .479 | .464 | .384 | .402 | .386 | .400 |
| ETTh1 | 192 | .424 | .425 | .426 | .431 | .445 | .441 | .433 | .419 | .446 | .437 | .441 | .436 | .437 | .424 | .460 | .445 | .427 | .425 | .471 | .474 | .525 | .492 | .436 | .429 | .437 | .432 |
| ETTh1 | 336 | .466 | .448 | .468 | .448 | .491 | .462 | .474 | .440 | .502 | .473 | .487 | .458 | .479 | .446 | .501 | .466 | .465 | .445 | .570 | .546 | .565 | .515 | .491 | .469 | .481 | .459 |
| ETTh1 | 720 | .459 | .466 | .464 | .467 | .506 | .497 | .473 | .462 | .517 | .494 | .509 | .494 | .481 | .470 | .500 | .488 | .472 | .468 | .461 | .621 | .594 | .558 | .521 | .500 | .519 | .516 |
| ETTh1 | Avg. | .430 | .434 | .434 | .436 | .457 | .452 | .441 | .428 | .463 | .452 | .457 | .449 | .446 | .434 | .469 | .454 | .437 | .434 | .529 | .522 | .541 | .507 | .458 | .450 | .456 | .452 |
| ETTh2 | 96 | .292 | .345 | .297 | .350 | .297 | .349 | .288 | .337 | .310 | .352 | .301 | .350 | .288 | .338 | .302 | .348 | .309 | .359 | .745 | .584 | .400 | .440 | .340 | .374 | .333 | .387 |
| ETTh2 | 192 | .371 | .397 | .372 | .397 | .378 | .399 | .374 | .392 | .390 | .403 | .381 | .399 | .374 | .390 | .388 | .400 | .390 | .406 | .877 | .656 | .528 | .509 | .402 | .414 | .477 | .476 |
| ETTh2 | 336 | .416 | .429 | .417 | .429 | .425 | .435 | .420 | .429 | .454 | .443 | .427 | .434 | .415 | .426 | .426 | .433 | .426 | .444 | 1.043 | .731 | .643 | .571 | .452 | .452 | .594 | .541 |
| ETTh2 | 720 | .422 | .443 | .425 | .445 | .432 | .448 | .422 | .439 | .420 | .437 | .430 | .446 | .420 | .440 | .431 | .446 | .445 | .464 | 1.104 | .763 | .874 | .679 | .462 | .468 | .831 | .657 |
| ETTh2 | Avg. | .376 | .403 | .378 | .405 | .383 | .408 | .376 | .398 | .394 | .409 | .384 | .407 | .374 | .398 | .393 | .407 | .393 | .418 | .942 | .684 | .611 | .550 | .414 | .427 | .559 | .515 |
| ETTm1 | 96 | .320 | .358 | .324 | .360 | .326 | .368 | .326 | .363 | .341 | .371 | .342 | .377 | .355 | .376 | .329 | .367 | .335 | .373 | .404 | .426 | .364 | .387 | .338 | .375 | .345 | .372 |
| ETTm1 | 192 | .366 | .383 | .368 | .385 | .378 | .393 | .366 | .386 | .380 | .390 | .383 | .396 | .391 | .392 | .367 | .385 | .372 | .390 | .450 | .451 | .398 | .404 | .374 | .387 | .380 | .389 |
| ETTm1 | 336 | .400 | .407 | .404 | .410 | .410 | .414 | .412 | .407 | .437 | .424 | .418 | .418 | .424 | .415 | .399 | .410 | .403 | .411 | .532 | .515 | .428 | .425 | .410 | .411 | .413 | .413 |
| ETTm1 | 720 | .467 | .445 | .472 | .445 | .474 | .451 | .468 | .441 | .507 | .462 | .487 | .450 | .487 | .450 | .454 | .439 | .461 | .442 | .666 | .589 | .487 | .461 | .478 | .450 | .474 | .453 |
| ETTm1 | Avg. | .387 | .398 | .392 | .400 | .398 | .407 | .393 | .399 | .416 | .412 | .408 | .412 | .414 | .407 | .387 | .400 | .393 | .404 | .513 | .496 | .419 | .419 | .400 | .406 | .403 | .407 |
| ETTm2 | 96 | .177 | .259 | .179 | .261 | .182 | .266 | .175 | .258 | .180 | .260 | .186 | .272 | .182 | .265 | .175 | .259 | .176 | .266 | .287 | .366 | .207 | .305 | .187 | .267 | .193 | .292 |
| ETTm2 | 192 | .244 | .306 | .244 | .304 | .252 | .313 | .241 | .300 | .243 | .303 | .254 | .314 | .246 | .304 | .241 | .302 | .240 | .307 | .414 | .492 | .290 | .364 | .249 | .309 | .284 | .362 |
| ETTm2 | 336 | .301 | .341 | .302 | .341 | .313 | .349 | .304 | .341 | .309 | .345 | .317 | .353 | .307 | .342 | .305 | .343 | .304 | .345 | .597 | .542 | .377 | .422 | .321 | .351 | .369 | .427 |
| ETTm2 | 720 | .399 | .398 | .401 | .400 | .416 | .409 | .406 | .400 | .407 | .403 | .412 | .407 | .407 | .398 | .402 | .400 | .406 | .400 | 1.730 | 1.042 | .558 | .524 | .408 | .403 | .554 | .522 |
| ETTm2 | Avg. | .280 | .326 | .281 | .326 | .290 | .333 | .281 | .326 | .285 | .328 | .293 | .337 | .286 | .327 | .281 | .326 | .282 | .330 | .757 | .610 | .358 | .404 | .291 | .333 | .350 | .401 |
| PEMS03 | 12 | .066 | .174 | .066 | .170 | .066 | .171 | - | - | .067 | .173 | .071 | .174 | .126 | .236 | .099 | .216 | - | - | .090 | .203 | .178 | .305 | .122 | .243 | .201 | .317 |
| PEMS03 | 24 | .090 | .199 | .092 | .202 | .088 | .197 | - | - | .095 | .206 | .097 | .208 | .246 | .334 | .142 | .259 | - | - | .121 | .240 | .257 | .371 | .118 | .223 | .201 | .317 |
| PEMS03 | 48 | .132 | .243 | .156 | .268 | .165 | .277 | - | - | .149 | .261 | .161 | .272 | .551 | .529 | .211 | .319 | - | - | .202 | .317 | .379 | .463 | .155 | .260 | .333 | .425 |
| PEMS03 | 96 | .192 | .297 | .231 | .334 | .213 | .313 | - | - | .239 | .341 | .240 | .338 | 1.057 | .787 | .269 | .370 | - | - | .262 | .367 | .490 | .539 | .228 | .317 | .457 | .515 |
| PEMS03 | Avg. | .120 | .228 | .136 | .243 | .133 | .240 | - | - | .138 | .245 | .142 | .248 | .495 | .472 | .180 | .291 | - | - | .169 | .281 | .326 | .419 | .147 | .248 | .278 | .375 |
| PEMS04 | 12 | .072 | .175 | .071 | .172 | .076 | .180 | - | - | .085 | .190 | .081 | .188 | .138 | .252 | .105 | .224 | - | - | .098 | .218 | .219 | .340 | .087 | .195 | .148 | .272 |
| PEMS04 | 24 | .084 | .190 | .090 | .199 | .084 | .192 | - | - | .118 | .226 | .099 | .211 | .258 | .348 | .153 | .275 | - | - | .131 | .256 | .292 | .398 | .103 | .215 | .224 | .340 |
| PEMS04 | 48 | .102 | .210 | .114 | .226 | .115 | .224 | - | - | .179 | .282 | .133 | .246 | .572 | .544 | .229 | .339 | - | - | .205 | .326 | .409 | .478 | .136 | .250 | .355 | .437 |
| PEMS04 | 96 | .127 | .233 | .137 | .248 | .137 | .248 | - | - | .303 | .382 | .172 | .283 | 1.137 | .820 | .291 | .389 | - | - | .402 | .457 | .492 | .532 | .190 | .303 | .452 | .504 |
| PEMS04 | Avg. | .099 | .203 | .103 | .211 | .103 | .211 | - | - | .171 | .270 | .121 | .232 | .526 | .491 | .195 | .307 | - | - | .209 | .314 | .353 | .437 | .129 | .241 | .295 | .388 |
| PEMS07 | 12 | .061 | .155 | .059 | .155 | .060 | .157 | - | - | .063 | .160 | .067 | .168 | .118 | .235 | .095 | .207 | - | - | .094 | .200 | .173 | .304 | .101 | .204 | .115 | .242 |
| PEMS07 | 24 | .076 | .173 | .078 | .178 | .082 | .184 | - | - | .090 | .192 | .088 | .190 | .242 | .341 | .150 | .262 | - | - | .139 | .247 | .271 | .383 | .101 | .204 | .210 | .329 |
| PEMS07 | 48 | .101 | .200 | .103 | .207 | .100 | .204 | - | - | .137 | .238 | .113 | .218 | .562 | .541 | .253 | .340 | - | - | .311 | .369 | .446 | .495 | .134 | .224 | .398 | .458 |
| PEMS07 | 96 | .120 | .221 | .123 | .224 | .117 | .218 | - | - | .208 | .304 | .172 | .283 | 1.096 | .795 | .346 | .404 | - | - | .396 | .442 | .628 | .577 | .181 | .279 | .594 | .553 |
| PEMS07 | Avg. | .089 | .187 | .091 | .191 | .090 | .191 | - | - | .125 | .224 | .102 | .205 | .504 | .478 | .211 | .303 | - | - | .235 | .315 | .380 | .440 | .124 | .225 | .329 | .395 |
| PEMS08 | 12 | .075 | .175 | .075 | .175 | .076 | .178 | - | - | .077 | .177 | .088 | .197 | .133 | .247 | .168 | .232 | - | - | .165 | .214 | .227 | .343 | .112 | .212 | .154 | .276 |
| PEMS08 | 24 | .094 | .196 | .103 | .203 | .110 | .216 | - | - | .113 | .213 | .138 | .243 | .249 | .343 | .224 | .281 | - | - | .215 | .260 | .318 | .409 | .141 | .238 | .248 | .353 |
| PEMS08 | 48 | .138 | .230 | .160 | .253 | .173 | .254 | - | - | .182 | .272 | .334 | .353 | .569 | .544 | .321 | .354 | - | - | .315 | .355 | .497 | .510 | .198 | .283 | .440 | .470 |
| PEMS08 | 96 | .226 | .299 | .252 | .314 | .271 | .321 | - | - | .308 | .354 | .458 | .436 | 1.166 | .814 | .408 | .417 | - | - | .377 | .397 | .721 | .592 | .320 | .351 | .674 | .565 |
| PEMS08 | Avg. | .133 | .225 | .148 | .236 | .157 | .242 | - | - | .170 | .254 | .254 | .306 | .529 | .487 | .280 | .321 | - | - | .268 | .307 | .441 | .464 | .193 | .271 | .379 | .416 |
| Exchange | 96 | .084 | .204 | .085 | .205 | .086 | .206 | .081 | .197 | .096 | .226 | .086 | .206 | .093 | .217 | .088 | .205 | .084 | .203 | .256 | .367 | .094 | .218 | .107 | .234 | .088 | .218 |
| Exchange | 192 | .178 | .301 | .178 | .301 | .181 | .303 | .172 | .295 | .222 | .353 | .177 | .299 | .184 | .307 | .176 | .299 | .171 | .294 | .470 | .509 | .184 | .307 | .226 | .344 | .176 | .315 |
| Exchange | 336 | .324 | .414 | .330 | .416 | .331 | .417 | .322 | .407 | .463 | .516 | .338 | .422 | .351 | .432 | .301 | .397 | .319 | .407 | 1.268 | .883 | .349 | .431 | .367 | .448 | .313 | .427 |
| Exchange | 720 | .846 | .693 | .852 | .696 | .858 | .699 | .846 | .693 | 1.251 | .856 | .847 | .691 | .886 | .714 | .901 | .714 | .805 | .677 | 1.767 | 1.068 | .852 | .698 | .964 | .746 | .839 | .695 |
| Exchange | Avg. | .358 | .403 | .361 | .404 | .364 | .407 | .355 | .398 | .508 | .488 | .368 | .409 | .378 | .417 | .367 | .404 | .345 | .395 | .940 | .707 | .370 | .413 | .416 | .443 | .354 | .414 |
| Weather | 96 | .155 | .205 | .157 | .207 | .165 | .209 | .164 | .211 | .174 | .213 | .174 | .213 | .192 | .232 | .177 | .218 | .159 | .218 | .158 | .230 | .202 | .261 | .172 | .220 | .196 | .255 |
| Weather | 192 | .208 | .250 | .209 | .252 | .215 | .255 | .213 | .254 | .228 | .261 | .224 | .258 | .240 | .271 | .225 | .259 | .211 | .266 | .273 | .305 | .242 | .298 | .219 | .261 | .237 | .296 |
| Weather | 336 | .260 | .282 | .263 | .292 | .273 | .296 | .271 | .295 | .281 | .299 | .281 | .298 | .292 | .307 | .278 | .297 | .267 | .310 | .273 | .335 | .287 | .335 | .280 | .306 | .283 | .335 |
| Weather | 720 | .351 | .346 | .352 | .345 | .353 | .349 | .352 | .347 | .357 | .349 | .359 | .351 | .364 | .353 | .354 | .348 | .352 | .362 | .398 | .418 | .351 | .386 | .365 | .359 | .345 | .381 |
| Weather | Avg. | .244 | .271 | .246 | .274 | .252 | .277 | .250 | .277 | .260 | .281 | .260 | .281 | .272 | .291 | .259 | .281 | .247 | .289 | .276 | .322 | .271 | .320 | .259 | .315 | .265 | .317 |
| Solar | 96 | .189 | .227 | .198 | .232 | .207 | .246 | .203 | .238 | .202 | .231 | .201 | .234 | .322 | .339 | .234 | .286 | - | - | .310 | .331 | .312 | .399 | .250 | .292 | .290 | .378 |
| Solar | 192 | .232 | .262 | .232 | .258 | .240 | .272 | .253 | .271 | .227 | .248 | .238 | .251 | .359 | .356 | .267 | .310 | - | - | .734 | .725 | .398 | .416 | .296 | .318 | .320 | .398 |
| Solar | 336 | .248 | .269 | .248 | .271 | .262 | .290 | .276 | .291 | .245 | .264 | .248 | .273 | .397 | .369 | .290 | .315 | - | - | .750 | .735 | .368 | .430 | .319 | .330 | .353 | .415 |
| Solar | 720 | .252 | .279 | .252 | .277 | .267 | .293 | .292 | .297 | .246 | .267 | .249 | .275 | .397 | .356 | .289 | .317 | - | - | .769 | .765 | .370 | .425 | .338 | .337 | .356 | .413 |
| Solar | Avg. | .230 | .259 | .233 | .259 | .244 | .275 | .256 | .274 | .230 | .253 | .234 | .261 | .369 | .356 | .270 | .307 | - | - | .641 | .639 | .347 | .417 | .301 | .319 | .330 | .401 |
| ECL | 96 | .134 | .231 | .130 | .226 | .139 | .237 | .143 | .240 | .144 | .236 | .148 | .240 | .201 | .281 | .181 | .270 | .173 | .275 | .219 | .314 | .237 | .329 | .168 | .272 | .197 | .282 |
| ECL | 192 | .155 | .248 | .156 | .249 | .165 | .261 | .158 | .255 | .161 | .252 | .167 | .258 | .201 | .283 | .188 | .274 | .195 | .288 | .231 | .322 | .236 | .330 | .184 | .289 | .196 | .285 |
| ECL | 336 | .171 | .267 | .171 | .264 | .177 | .274 | .171 | .268 | .174 | .267 | .179 | .272 | .215 | .298 | .204 | .293 | .206 | .300 | .246 | .337 | .249 | .344 | .198 | .300 | .209 | .301 |
| ECL | 720 | .195 | .289 | .195 | .287 | .214 | .304 | .198 | .293 | .204 | .294 | .220 | .310 | .257 | .331 | .246 | .324 | .231 | .335 | .280 | .363 | .284 | .373 | .220 | .320 | .245 | .333 |
| ECL | Avg. | .163 | .259 | .163 | .256 | .174 | .269 | .168 | .264 | .171 | .262 | .179 | .270 | .219 | .298 | .205 | .290 | .201 | .300 | .244 | .334 | .251 | .344 | .192 | .295 | .212 | .300 |
| Traffic | 96 | .375 | .256 | .351 | .249 | .379 | .260 | .444 | .285 | .400 | .267 | .395 | .268 | .649 | .389 | .462 | .295 | .570 | .310 | .522 | .290 | .805 | .493 | .593 | .321 | .650 | .396 |
| Traffic | 192 | .387 | .263 | .389 | .260 | .409 | .272 | .455 | .289 | .400 | .264 | .417 | .277 | .601 | .366 | .466 | .296 | .577 | .321 | .530 | .293 | .756 | .474 | .617 | .336 | .598 | .370 |
| Traffic | 336 | .401 | .272 | .415 | .277 | .418 | .277 | .471 | .299 | .416 | .271 | .433 | .283 | .609 | .369 | .482 | .304 | .588 | .324 | .558 | .305 | .762 | .477 | .629 | .336 | .605 | .373 |
| Traffic | 720 | .415 | .284 | .443 | .290 | .461 | .297 | .511 | .320 | .434 | .284 | .467 | .300 | .647 | .387 | .514 | .322 | .597 | .337 | .589 | .328 | .719 | .449 | .640 | .350 | .645 | .394 |
| Traffic | Avg. | .394 | .269 | .399 | .269 | .417 | .277 | .470 | .298 | .413 | .272 | .428 | .282 | .626 | .378 | .481 | .304 | .583 | .323 | .550 | .304 | .760 | .473 | .620 | .336 | .625 | .383 |
| 1st Count | | 29 | 25 | 9 | 11 | 4 | 4 | 6 | 11 | 2 | 2 | 0 | 0 | 3 | 3 | 5 | 2 | 3 | 3 | 1 | 0 | 0 | 0 | 0 | 0 | 2 | 0 |
| 2nd Count | | 18 | 16 | 24 | 22 | 4 | 5 | 4 | 7 | 2 | 1 | 4 | 2 | 0 | 6 | 0 | 4 | 2 | 2 | 0 | 0 | 0 | 0 | 0 | 0 | 2 | 0 |

Table F.1: **Full results of TS forecasting tasks.** For fair comparison, we adopt the CD architecture for Minusformer across all datasets, as it consistently outperforms the CI architecture on average.

## G PSEUDOCODE OF CSM

Algorithm 2 shows the pseudocode for the proposed pretraining task, channel similarity modeling (CSM), where an arbitrary TS encoder can be employed.

---

**Algorithm 2** Channel Similarity Modeling (CSM)

---

**Input**: $\mathbf{X} = [\mathbf{X}_1, \ldots, \mathbf{X}_L] : (B, L, C)$

1: $\mathbf{R_X} : (B, C, C) \leftarrow$ Calculate correlation matrix with $\mathbf{X}$
2: $\mathbf{Z} : (B, C, D) \leftarrow$ Encoder($\mathbf{X}$)
3: $\mathbf{R_Z} : (B, C, C) \leftarrow$ Calculate correlation matrix with $\mathbf{Z}$
4: Minimize $d_{\text{CSM}}(\mathbf{R_X}, \mathbf{R_Z})$

---

## H CHANNEL ORDERS FOR TWO VIEWS

Table H.1 shows the results with the average MSE across four horizons, indicating that fixing the order yields better performance than permuting the order, especially with a large number of channels ($C \geq 100$). Note that we use the same model hyperparameters as S-Mamba (Wang et al., 2025) for all settings in Table H.1.

| $F$: Fixed , $R$: Random , $X^\star$: Reverse of $X$ | | | | | | Impr. (Robust.) |
|---|---|---|---|---|---|---|
| Order | $\mathbf{z}_1$ | $F$ | $F$ | $R_1$ | $R$ | |
| | $\mathbf{z}_2$ | $F^\star$ | $R$ | $R_2$ | $R^\star$ | |
| Dataset | $C$ | (a) | (b) | (c) | (d) | (d) $\rightarrow$ (a) |
| ETTh1 | 7 | **.442** | .443 | .446 | .443 | 0.2% |
| ETTh2 | 7 | **.382** | **.382** | **.382** | **.382** | 0.0% |
| ETTm1 | 7 | **.396** | **.396** | **.396** | **.396** | 0.0% |
| ETTm2 | 7 | **.284** | .285 | .285 | .285 | 0.4% |
| Exchange | 8 | **.363** | .364 | .365 | .364 | 0.3% |
| Weather | 21 | **.257** | .258 | .260 | .260 | 1.2% |
| Average | | **.354** | .355 | .356 | .355 | 0.3% |
| Solar | 137 | **.242** | .245 | .245 | .246 | 1.6% |
| PEMS03 | 358 | **.137** | .144 | .150 | .151 | 9.3% |
| PEMS04 | 307 | **.107** | .112 | .116 | .117 | 8.5% |
| PEMS07 | 883 | **.091** | .096 | .097 | .096 | 5.2% |
| PEMS08 | 170 | **.162** | .163 | .169 | .172 | 5.8% |
| ECL | 321 | **.169** | .174 | .181 | .183 | 7.7% |
| Traffic | 862 | **.412** | .422 | .423 | .423 | 2.6% |
| Average | | **.189** | .194 | .197 | .198 | 4.9% |

(Row group labels: $C < 100$ for the first block, $C \geq 100$ for the second block.)

Table H.1: Channel orders for two views.

Figure H.1 illustrates the four candidates for generating two embedding vectors, $\mathbf{z}_1$ and $\mathbf{z}_2$, for regularization, based on whether the channel order is fixed or randomly permuted in each iteration. Results in Table H.1 indicate that fixing the order during training yields the best performance, with performance degrading as the order becomes random, especially with many channels, though it remains robust with fewer channels. We argue that a fixed order is preferable due to the instability introduced by randomness during training, as shown in Figure H.1, which displays the training loss for two datasets (Zhou et al., 2021; Liu et al., 2022) with varying numbers of channels.

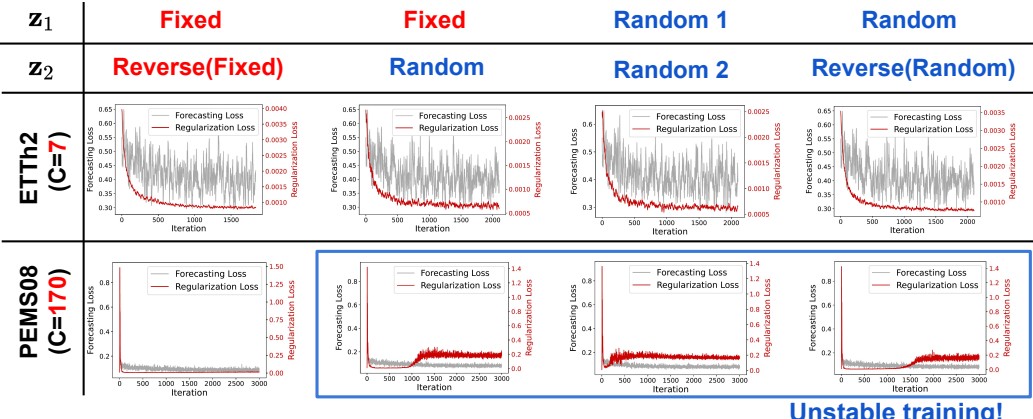

Figure H.1: Fixed vs. random order for generating two views, $\mathbf{z}_1$ and $\mathbf{z}_2$.

## I ROBUSTNESS TO DISTANCE METRIC

To assess whether FSMamba is sensitive to the choice of distance metrics $d$ for the regularization term and $d_{\text{CSM}}$ for CSM when comparing the two matrices, we compare various metrics, including (negative) cosine similarity, $\ell_1$ loss, and $\ell_2$ loss. Tables I.1 and I.2 show the average MSE across four different horizons for the distance metric used in the regularization term and CSM, respectively, demonstrating that the performance is robust to the choice of distance metric, where we choose $\ell_2$ loss throughout the experiment for both metrics.

| Dataset | FSMamba (SL) | | | S-Mamba |
|---|---|---|---|---|
| | Cosine | $\ell_1$ Loss | $\ell_2$ Loss | |
| ETTh1 | **.434** | **.434** | **.434** | .457 |
| ETTh2 | **.378** | **.378** | **.378** | .383 |
| ETTm1 | .393 | **.392** | **.392** | .398 |
| ETTm2 | **.281** | **.281** | **.281** | .290 |
| PEMS03 | .137 | .138 | .136 | **.133** |
| PEMS04 | .104 | **.103** | **.103** | **.103** |
| PEMS07 | **.090** | .091 | **.090** | **.090** |
| PEMS08 | .148 | **.146** | .148 | .157 |
| Exchange | **.361** | .362 | **.361** | .364 |
| Weather | **.245** | **.245** | .247 | .252 |
| Solar | **.233** | **.233** | **.233** | .244 |
| ECL | .164 | **.163** | **.163** | .174 |
| Traffic | .402 | .400 | **.399** | .417 |
| Average | **.259** | **.259** | **.259** | .266 |

Table I.1: Robustness to $d$ for regularization.

| Dataset | FSMamba (SSL) | | S-Mamba |
|---|---|---|---|
| | $\ell_1$ Loss | $\ell_2$ Loss | |
| ETTh1 | **.430** | **.430** | .457 |
| ETTh2 | .377 | **.376** | .383 |
| ETTm1 | **.387** | **.387** | .398 |
| ETTm2 | **.280** | **.280** | .290 |
| PEMS03 | .121 | **.120** | .133 |
| PEMS04 | **.099** | **.099** | .103 |
| PEMS07 | **.089** | **.089** | .090 |
| PEMS08 | .135 | **.133** | .157 |
| Exchange | **.358** | **.358** | .364 |
| Weather | .247 | **.244** | .252 |
| Solar | .232 | **.230** | .244 |
| ECL | .166 | **.163** | .174 |
| Traffic | .395 | **.394** | .417 |
| Average | .258 | **.257** | .266 |

Table I.2: Robustness to $d_{\text{CSM}}$ for CSM.

## J    COMPARISON OF GPU MEMORY USAGE

Figure J.1 visualizes GPU memory usage by dataset and method, demonstrating that our method is more efficient than both S-Mamba (Wang et al., 2025) and iTransformer (Liu et al., 2024a). Specifically, Mamba-based methods are more efficient than Transformer-based methods when $C$ is large, as Mamba has nearly-linear complexity, whereas Transformers have quadratic complexity.

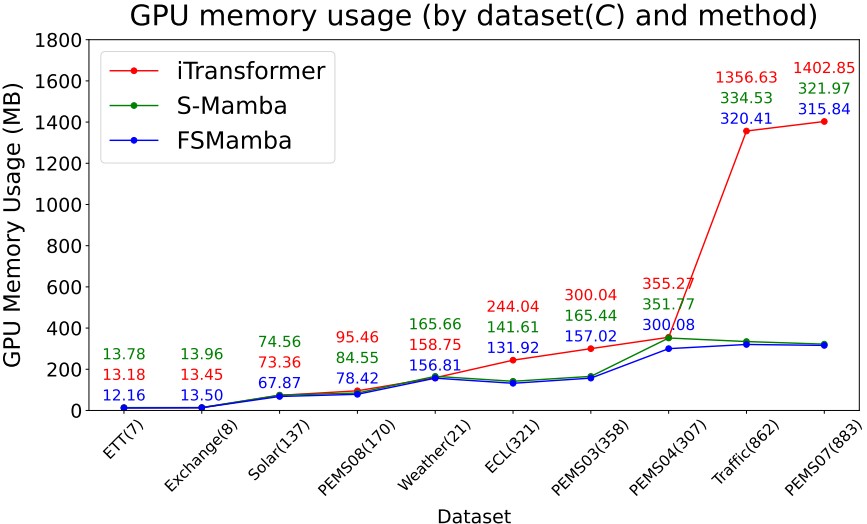

Figure J.1: Comparison of GPU memory usage.

## K    APPLICATION OF CSM TO ITRANSFORMER

To demonstrate the effectiveness of CSM, we apply it to iTransformer(Liu et al., 2024a), a representative Transformer-based model for capturing CD, across three datasets (ETTh1 (Zhou et al., 2021), Weather (Wu et al., 2021), ECL (Wu et al., 2021)) with varying numbers of channels. The results are shown in Table K.1, where CSM consistently enhances iTransformer across different datasets and forecast horizons.

| | ETTh1 ($C = 7$) | | | | Weather ($C = 21$) | | | | ECL ($C = 321$) | | | |
| | - | | + CSM | | - | | + CSM | | - | | + CSM | |
| $H$ | MSE | MAE | MSE | MAE | MSE | MAE | MSE | MAE | MSE | MAE | MSE | MAE |
|---|---|---|---|---|---|---|---|---|---|---|---|---|
| 96 | .387 | .405 | **.385** | **.401** | **.174** | .215 | **.174** | **.210** | .148 | .240 | **.145** | **.237** |
| 192 | **.441** | **.436** | **.441** | **.436** | .224 | .258 | **.221** | **.255** | .167 | .258 | **.163** | **.252** |
| 336 | .487 | .458 | **.484** | **.453** | .281 | .298 | **.280** | **.296** | .179 | .272 | **.174** | **.266** |
| 720 | .509 | .494 | **.491** | **.487** | .359 | .351 | **357** | **.348** | .220 | .310 | **.215** | **.304** |

Table K.1: Application of CSM to iTransformer across three datasets with various channel counts.

## L APPLICATION TO HIGH-DIMENSIONAL DATA

We conduct experiments on two high-dimensional datasets, following the previous work (Ni et al., 2025), as follows:

- [1] **M5 Forecasting** (Howard et al., 2020): A high-dimensional retail sales dataset from Walmart containing 30,490 item–store combinations across 1,941 days. Due to inherent sparsity, we aggregate sales by summing each item's sales across all departments and stores, reducing the dataset to 3,049 aggregated items.
- [2] **S&P 500 Index**: The S&P 500 is a stock market index, consisting of 503 common stocks issued by 500 large-cap companies traded on American stock exchanges. We use yfinance[3], a Python package for financial data retrieval, to download market data for these stocks from Yahoo Finance. We select the latest S&P 500 company list (as of 03/14/2025) and extract daily market data spanning the past 30 years (7,553 days). To ensure data consistency, we retain only the companies that were publicly traded 30 years ago. For each company, we extract five key market variables: Open, Close, High, Low, and Volume, resulting in a total of 1,475 dimensions.

We set the input horizon ($L$) and forecast horizon ($H$) to (96, 28) for M5 and (21, 7) for S&P 500, respectively, with a 70/10/20 train/validation/test split for both datasets. Table L.2 compares the results of S-Mamba (Wang et al., 2025) (bidirectional Mamba) and FSMamba (unidirectional Mamba with regularization), demonstrating the effectiveness of our method on high-dimensional datasets, as it achieves competitive performance compared to S-Mamba, which leverages bidirectional modeling.

| Settings | M5 | S&P 500 |
|---|---|---|
| $L$ | 96 | 21 |
| $H$ | 28 | 7 |
| $C$ | 3,049 | 1,475 |
| Frequency | Daily | Daily |

Table L.1: High-dimensional data statistics.

| | S-Mamba | | FSMamba | |
|---|---|---|---|---|
| | MSE | MAE | MSE | MAE |
| M5 Forecasting | .374 | .872 | **.372** | **.865** |
| S&P 500 Index | .399 | .267 | **.395** | **.265** |

Table L.2: TS forecasting results on high-dimensional datasets.

---

[3]https://github.com/ranaroussi/yfinance

# M    SCAN DIRECTION STABILITY EXPERIMENT

To validate our hypothesis that scan-order consistency is critical for training stability in Mamba-based models, we conducted a controlled experiment comparing two channel scanning strategies: (1) **reverse**, where the channel order is deterministically reversed (fixed scan path), and (2) **random**, where channels are randomly permuted at every training step. We trained a minimal Mamba model (128 hidden units, 2 layers) on synthetic multivariate TS with varying channel counts ($C \in \{5, 30, 50, 100, 200\}$) for 10 epochs (200 steps per epoch, batch size 64). The synthetic data consists of 10,000 samples with lookback length $L = 32$ and forecast horizon $H = 8$, generated from 4 latent sinusoidal drivers with Gaussian noise.

| Hyperparameter | Value |
|---|---|
| Channel counts ($C$) | {5, 30, 50, 100, 200} |
| Lookback length ($L$) | 32 |
| Forecast horizon ($H$) | 8 |
| Training samples | 10,000 |
| Batch size | 64 |
| Epochs | 10 |
| Steps per epoch | 200 |
| Learning rate | 0.002 |
| Hidden dimension | 128 |
| Model layers | 2 |

Table M.1: Experimental setup.

Figure M.1 demonstrates that random channel permutation leads to substantially higher training instability compared to fixed reverse scanning, with the gap widening as dimensionality increases. This validates that consistent scan-order is critical for stable Mamba training, as random permutations disrupt the deterministic recurrent state evolution required for convergence.

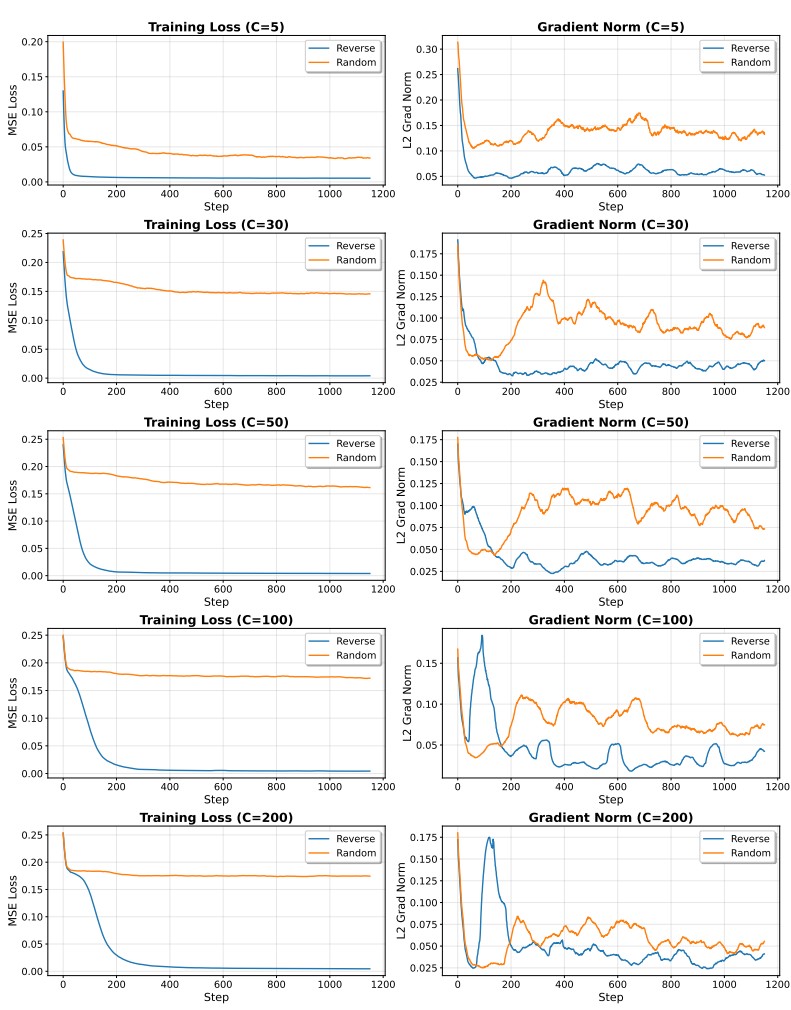

Figure M.1: Scan direction stability experiments across various channel counts.

