# OpenReview forum: "Is Bidirectionality Necessary in Mamba for Time Series Forecasting?"
_ICLR.cc/2026/Conference — Submitted to ICLR 2026_

### Official Review · Reviewer_P4ub · 2025-10-30

**Soundness:** 3
**Presentation:** 4
**Contribution:** 2
**Rating:** 4
**Confidence:** 4

**Summary:**

The paper studies whether bidirectionality is necessary for Mamba in multivariate time series forecasting. Since channels lack natural order, applying bidirectional Mamba for channel dependencies (CD) adds complexity. The authors propose FSMamba, a unidirectional model with a regularization term that minimizes the distance between embeddings of reversed channel orders and a channel similarity modeling (CSM) pretraining step. Experiments on multiple datasets show FSMamba achieves comparable or better performance than bidirectional models with fewer parameters, proving that bidirectionality is unnecessary for effective CD modeling.

**Strengths:**

1. Clear and easy-to-understand writing with clean and visually appealing figures.
2. Detailed experimental analysis

**Weaknesses:**

1. Since Mamba itself is designed for TD, the motivation for applying it to CD is not well justified.
2. The evaluation of Mamba’s modeling capability on CD is insufficient.

**Questions:**

1. The paper proposes a regularization method that minimizes the distance between two embedding vectors generated with reversed channel orders. However, this seems to prevent Mamba, as a sequence modeling method, from capturing sequential information. I do not understand the rationale behind this design. The final architecture appears equivalent to using Selective SSM to capture CD. Please explain why this design is necessary and why it works.

2. Since the existing benchmarks contain too few dimensions, to further validate FSMamba’s capability in modeling CD, it would be helpful to include experiments on some datasets from Time-HD benchmark [1] and Wike2000 from TFB [2].

[1] Ni, J., Wang, S., Liu, Z., Shi, X., Zhong, X., Ye, Z., & Jin, W. (2025). U-Cast: Learning Hierarchical Structures for High-Dimensional Time Series Forecasting. arXiv preprint arXiv:2507.15119.
[2] Qiu, X., Hu, J., Zhou, L., Wu, X., Du, J., Zhang, B., ... & Yang, B. (2024). Tfb: Towards comprehensive and fair benchmarking of time series forecasting methods. arXiv preprint arXiv:2403.20150

---

> ### Author Response · Authors · 2025-11-21
>
> > [W1] Since Mamba itself is **designed for “TD”**, the motivation for **applying it to “CD”** is not well justified.
>
> We agree with the reviewer that **Mamba can be thought to be designed for TD**, as it is an **”SSM”-based architecture** that processes inputs sequentially. However, we believe that ***its essence does not lie in temporal modeling itself***, but rather in its ability to **efficiently replace** the **(quadratic-complexity) attention mechanism** with a **(linear-complexity) selective-scan mechanism** for modeling token dependencies.
>
>
> This property allows Mamba to be effectively extended ***beyond temporal domains***  as an **alternative to Transformer**,  and has been successfully applied to various **non-sequential** tasks—such as tabular data [A, B] and channel dependency modeling in TS [C–J]—where its efficiency and dependency-capturing capability remain advantageous. In particular, both prior studies [J] and our results (**Table 7**) consistently show that ***Mamba achieves stronger performance when applied to CD than TD***, reinforcing its effectiveness beyond TD.
>
>
> Furthermore, this point is also **acknowledged in our paper**:
>
>
> - 1. **[L47–50]** As noted in L47–50, we explicitly state:  “~ Mamba is an SSM-based model designed for sequential inputs, making it more natural to capture TD rather than CD. Nonetheless, we focus on Mamba capturing CD instead of TD, following recent works ~ due to their superior performance.”
> - 2. **[Table 4]** Table 4 shows that various methods leverage Mamba **beyond TD** to effectively capture CD.
>
>
> A detailed explanation of why applying a **sequential architecture** to a **non-sequential domain** is largely a **superficial difference**, and that the true essence of Mamba lies in replacing the ***”quadratic-complexity” attention*** of Transformers with a ***”linear-complexity” selective-scan mechanism***, can be found in our response to **[Q1]** below.
>
>
> Please feel free to reach out with any further questions regarding this point!
>
>
> &nbsp;
>
> - [A] Thielmann, Anton Frederik, et al. "Mambular: A sequential model for tabular deep learning." arXiv 2024.
> - [B] Ahamed, Md Atik, and Qiang Cheng. "MambaTab: A plug-and-play model for learning tabular data." 2024 IEEE 7th International Conference on Multimedia Information Processing and Retrieval (MIPR). IEEE, 2024.
> - [C] Aobo Liang, Xingguo Jiang, Yan Sun, and Chang Lu. Bi-mamba+: Bidirectional mamba for time series forecasting. arXiv 2024.
> - [D] ​​Zixuan Weng, Jindong Han, Wenzhao Jiang, and Hao Liu. Simplified mamba with disentangled dependency encoding for long-term time series forecasting. arXiv 2024.
> - [E] Ali Behrouz, Michele Santacatterina, and Ramin Zabih. Mambamixer: Efficient selective state space models with dual token and channel selection. arXiv 2024.
> - [F] Md Atik Ahamed and Qiang Cheng. Timemachine: A time series is worth 4 mambas for long-term forecasting. In ECAI, 2024.
> - [G] Xiuding Cai, Yaoyao Zhu, Xueyao Wang, and Yu Yao. Mambats: Improved selective state space models for long-term time series forecasting. arXiv 2024.
> - [H] Shusen Ma, Yu Kang, Peng Bai, and Yun-Bo Zhao. Fmamba: Mamba based on fast-attention for multivariate time-series forecasting. arXiv 2024.
> - [I] Sijie Xiong, Shuqing Liu, Cheng Tang, Fumiya Okubo, Haoling Xiong, and Atsushi Shimada. Attention mamba: Time series modeling with adaptive pooling acceleration and receptive field enhancements. arXiv 2025.
> - [J] Zihan Wang, Fanheng Kong, Shi Feng, Ming Wang, Xiaocui Yang, Han Zhao, Daling Wang, and Yifei Zhang. Is mamba effective for time series forecasting? Neurocomputing 2025.

---

> ### Author Response · Authors · 2025-11-21
>
> > [W2] The evaluation of **Mamba’s modeling capability on CD** is insufficient.
> We believe our evaluation of **Mamba’s capability to model CD** is sufficiently supported by the following experiments:
> - **[Table 7]** compares **MLP, Mamba, and Transformer** architectures—each tailored to capture either TD or CD—across **13 datasets** and **4 forecasting horizons**, demonstrating Mamba’s effectiveness in modeling CD.
> - **[Table 4]** further reports results from various architectures employing Mamba for TD/CD modeling, where representative **Mamba-based CD models** (e.g., Bi-Mamba+, S-Mamba, FSMamba (Ours)) consistently achieve strong performance.
>
> Furthermore, **Mamba’s capability in capturing CD** has also been demonstrated in several prior works (**[A–H]**).
>
> Nonetheless, if the reviewer still finds any aspect insufficient, we would be happy to include additional experiments to further reinforce our analysis!
>
> &nbsp;
>
> - [A] Aobo Liang, Xingguo Jiang, Yan Sun, and Chang Lu. Bi-mamba+: Bidirectional mamba for time series forecasting. arXiv 2024.
> - [B] ​​Zixuan Weng, Jindong Han, Wenzhao Jiang, and Hao Liu. Simplified mamba with disentangled dependency encoding for long-term time series forecasting. arXiv 2024.
> - [C] Ali Behrouz, Michele Santacatterina, and Ramin Zabih. Mambamixer: Efficient selective state space models with dual token and channel selection. arXiv 2024.
> - [D] Md Atik Ahamed and Qiang Cheng. Timemachine: A time series is worth 4 mambas for long-term forecasting. In ECAI, 2024.
> - [E] Xiuding Cai, Yaoyao Zhu, Xueyao Wang, and Yu Yao. Mambats: Improved selective state space models for long-term time series forecasting. arXiv 2024.
> - [F] Shusen Ma, Yu Kang, Peng Bai, and Yun-Bo Zhao. Fmamba: Mamba based on fast-attention for multivariate time-series forecasting. arXiv 2024.
> - [G] Sijie Xiong, Shuqing Liu, Cheng Tang, Fumiya Okubo, Haoling Xiong, and Atsushi Shimada. Attention mamba: Time series modeling with adaptive pooling acceleration and receptive field enhancements. arXiv 2025.
> - [H] Zihan Wang, Fanheng Kong, Shi Feng, Ming Wang, Xiaocui Yang, Han Zhao, Daling Wang, and Yifei Zhang. Is mamba effective for time series forecasting? Neurocomputing 2025.
>
>
> &nbsp;
>
> ---
> &nbsp;
>
> > [Q1] The paper proposes a regularization method that minimizes the distance between two embedding vectors generated with reversed channel orders. However, this seems to prevent Mamba, as a **”sequence” modeling** method, from capturing **”sequential” information**. I do not understand the rationale behind this design. The final architecture appears equivalent to using Selective SSM **to capture CD**. Please explain why this design is necessary and why it works.
>
>
> We believe that the essence of Mamba’s core SSSM lies ***not in its sequential nature***, but in its ability to **efficiently replace** the ***(quadratic-complexity) attention mechanism*** with a ***(linear-complexity) selective-scan mechanism***.
>
>
> As shown in **Figure E.1**, a Mamba block combines 1) a lightweight 1D convolution for local context with the **2) SSSM**, the key module enabling **”linear-complexity” token dependency modeling**. Our regularization preserves this essence — it maintains the **selective-scan mechanism** while removing the bias from fixed scan order, enabling Mamba to capture CD effectively.
>
>
> Regarding the concern about applying Mamba’s **sequential** design to **non-sequential** data, we believe this issue is largely superficial. Just as (non-sequential) **Transformers** became suitable for **sequential** tasks like natural language after introducing **positional encodings**, Mamba’s sequential property merely arises from its scan direction. By regularizing it, we remove that sequential order bias while retaining Mamba’s original efficiency and design principles.
>
>
> For clarity, the originality of our approach can be summarized as follows:
> | Model | Bias | Capture Token Dependencies | Complexity |
> |--------|------|-----------------------------|-------------|
> | Transformer | ✗ | ✓ | Quadratic |
> | + Positional Encoding | Adds bias | ✓ | Quadratic |
> | Mamba | ✓ | ✓ | Linear |
> | + Regularization (Ours) | **Reduces bias** | ✓ | Linear |
>
>
> In short, our method retains Mamba’s core strength—**efficient dependency modeling**—while reducing its inherent directional bias, making it more suitable for capturing CD rather than enforcing TD.
>
>
> Moreover, our **extensive experimental results** across **13 datasets and 4 horizons** consistently support this rationale, empirically confirming the validity and effectiveness of the proposed design.

---

> ### Author Response · Authors · 2025-11-21
>
> > [Q2] Since the existing benchmarks contain **too few dimensions**, to further validate FSMamba’s capability in modeling CD, it would be helpful to include experiments on some datasets from Time-HD benchmark [1] and Wike2000 from TFB [2].
>
>
> The **Traffic** dataset used in our experiments already includes **”862 channels”**, which is generally considered a **high-dimensional** multivariate TS benchmark. Note that the paper mentioned by the reviewer (*U-Cast: Learning Hierarchical Structures for High-Dimensional Time Series Forecasting* [A]) was released on **arXiv after our submission (2025-09-24)** and specifically addresses the newly proposed High-Dimensional Time Series Forecasting (HDTSF) setting.
>
>
> Nevertheless, following the reviewer’s feedback, we have **added experiments on two additional high-dimensional datasets** — **M5 (3049 channels)** and **S&P 500 (1475 channels)** — to further evaluate our method’s effectiveness on high-dimensional datasets. The datasets statistics and the results are shown below:
>
>
> &nbsp;
>
> | **Settings** | **M5** | **S&P 500** |
> |:-------------:|:------:|:------------:|
> | $L$ | 96 | 21 |
> | $H$ | 28 | 7 |
> | $C$ | **3,049** | **1,475** |
> | Frequency | Daily | Daily |
>
> &nbsp;
>
> | **Dataset** | **S-Mamba** |      | **FSMamba** |      |
> |:-------------|:-----------:|:----:|:------------:|:----:|
> |              | MSE | MAE | MSE | MAE |
> | **M5 Forecasting** | 0.374 | 0.872 | **0.372** | **0.865** |
> | **S&P 500 Index** | 0.399 | 0.267 | **0.395** | **0.265** |
>
>
> These results are now included in the revised version (**[Appendix L] Application to High-dimensional datasets**). If there are any additional datasets or aspects you would like us to further examine, please feel free to let us know!
>
> &nbsp;
>
>
> **Please let us know if there are any remaining issues you'd like to discuss!**

---

### Official Review · Reviewer_7wxw · 2025-10-31

**Soundness:** 2
**Presentation:** 3
**Contribution:** 2
**Rating:** 4
**Confidence:** 4

**Summary:**

This paper proposes FSMamba, a unidirectional Mamba model for time series forecasting that addresses the "sequential order bias" found in bidirectional models. It uses a "Flipped Siamese" regularization strategy to enforce robustness to channel order and removes the 1D-convolution layer from the Mamba block, arguing it is unsuited for non-sequential channel data. The authors also introduce Channel Similarity Modeling (CSM), a pretraining task designed to preserve channel correlations from the data space to the latent space.

**Strengths:**

1. Principled Bias Correction: The paper clearly identifies the sequential order bias from applying Mamba to non-sequential channels. The proposed "Flipped Siamese" regularization is an intuitive and effective method to enforce order robustness using a single, shared-weight unidirectional model, avoiding the inefficiency of two-model bidirectional approaches.
2. Computational Efficiency: FSMamba achieves state-of-the-art performance while being significantly more efficient than bidirectional baselines like S-Mamba. It uses 37.6%–38.1% fewer parameters, consumes less GPU memory, and demonstrates faster training and inference times.
3. Thorough Validation: The method is validated on 13 diverse datasets against numerous strong baselines. The paper includes extensive ablation studies (Section 6) that confirm the benefits of the regularization, 1D-conv removal, and CSM pretraining.

**Weaknesses:**

1. Limited Permutation Robustness: The regularization strategy only enforces robustness against a single permutation (the reversed channel order) not general permutation invariance. The paper dismisses using random permutations as "unstable" without a deep investigation, meaning the model is order-robust only to a single, specific flip.
2. Disconnected Pretraining Task: The Channel Similarity Modeling (CSM) pretraining task, which aims to preserve channel correlations, feels disconnected from the paper's main thesis on sequential order bias and its novelty is debatable.
3. Contradictory 1D-Convolution Results: The paper justifies removing the 1D-convolution by stating channels lack sequential order. However, it also notes PEMS datasets do have a meaningful geographical order. The results in Table 9 show removing the 1D-conv does not harm performance on PEMS, a counter-intuitive finding that is not adequately explained and weakens the motivation for the removal.

**Questions:**

1. Why random permutation result in unstable training? Why can this strategy not lead to better robustness compared to the reverse flip?
2. Does the conclusion in Fig. 7  and the robustness generalize to even higher-dimensional data[1]?
3. For the CSM pretraining task, why only preserving the linear correlations? Will preserving non-linear correlations also help?
4. Can the two loss strategies (L_reg and L_CSM) be applied to other CD models like iTransformer[2] and Duet[3]?


References:
[1] Ni J, Wang S, Liu Z, Shi X, Zhong X, Ye Z, Jin W. U-Cast: Learning Hierarchical Structures for High-Dimensional Time Series Forecasting. arXiv preprint arXiv:2507.15119. 2025 Jul 20.
[2] Liu Y, Hu T, Zhang H, Wu H, Wang S, Ma L, Long M. itransformer: Inverted transformers are effective for time series forecasting. arXiv preprint arXiv:2310.06625. 2023 Oct 10.
[3]Qiu X, Wu X, Lin Y, Guo C, Hu J, Yang B. Duet: Dual clustering enhanced multivariate time series forecasting. InProceedings of the 31st ACM SIGKDD Conference on Knowledge Discovery and Data Mining V. 1 2025 Jul 20 (pp. 1185-1196).

---

> ### Author Response · Authors · 2025-11-21
>
> > [W] Limited Permutation Robustness: The regularization strategy **only enforces robustness against a single permutation (the reversed channel order) not general permutation invariance**. The paper dismisses using random permutations as **"unstable"** without a deep investigation, meaning the model is order-robust only to a single, specific flip.
>
>
> > [Q1] Why random permutation result in **unstable training**? Why can this strategy not lead to better robustness compared to the reverse flip?
>
>
> We would like to clarify that our method is ***not confined to a single (reversed) permutation***, but rather designed for **efficient and stable learning of permutation robustness**. The **reversed order** is selected as the **most symmetric and learnable configuration**, providing an efficient way to encourage invariance without introducing unnecessary instability.
>
>
> While achieving full **general permutation invariance** might be possible, we believe it is **difficult to achieve by learning in practice**, as training under random or continuously changing channel orders significantly increases optimization complexity and instability. This is empirically shown in **Figure H.1**, where random permutations cause fluctuating losses and hinder convergence, especially in high-dimensional settings.
>
>
> In contrast, using a **fixed reversed order** offers a balanced trade-off: it allows the model to learn direction-invariant representations while maintaining stable optimization. As evidenced in **Table 12**, FSMamba also maintains robustness across multiple unseen random permutations, confirming that it **generalizes beyond the specific reversed case**.
>
>
> &nbsp;
>
> Furthermore, to address the reviewer’s concern regarding ***why random permutations result in unstable training***, we conducted an additional analysis described in **[Appendix M]: Scan Direction Stability Experiment**. In this experiment, we trained a minimal Mamba-based model on synthetic multivariate time series with varying channel counts (C ∈ {5, 30, 50, 100, 200}) under two scanning strategies: (1) **reverse** (fixed inverted order, consistent scan path) and (2) **random** (channel permutation changes every batch). The results demonstrate that ***random permutation leads to significantly higher loss variance*** (e.g., std = 0.0047 vs. 0.0002 for C=30) and gradient instability (std = 0.0285 vs. 0.0143), with the gap widening as dimensionality increases. This confirms that maintaining a deterministic scan direction is crucial for stable recurrent state propagation in Mamba architectures.
>
>
> Please refer to **[Appendix M] Scan Direction Stability Experiment** of the revised version for full experimental details.
>
>
> &nbsp;
>
> ---
> &nbsp;
>
>
> > [W2] **Disconnected Pretraining Task**: The Channel Similarity Modeling (CSM) pretraining task, which aims to preserve channel correlations, feels disconnected from the paper's main thesis on sequential order bias and its novelty is debatable.
>
>
> Thank you for the feedback. While we acknowledge that **CSM primarily enhances CD modeling** and is **not the main source of novelty** compared to our proposed regularization, it remains conceptually connected to our study. Since **Mamba inherently struggles to capture CD due to its sequential design**, we introduce **1) the regularization** to mitigate order bias and **2) CSM pretraining** to enhance CD modeling, thereby addressing complementary aspects of Mamba’s CD limitations.
>
>
> Nonetheless, if its placement feels disconnected, we are open to **moving it to the Appendix** or treating it as an auxiliary study in the revision.

---

> ### Author Response · Authors · 2025-11-21
>
> > [W3] **Contradictory 1D-Convolution Results:** The paper justifies removing the 1D-convolution by stating channels lack sequential order. However, it also notes PEMS datasets do have a meaningful geographical order. The results in Table 9 show removing the 1D-conv does not harm performance on PEMS, a counter-intuitive finding that is not adequately explained and weakens the motivation for the removal.
>
>
> We believe the results in **Table 9** are not counter-intuitive for the following reasons:
> - (a) Datasets without channel order (e.g., ETT, Weather):
>  Removing the 1D-conv **improves** performance, as the channels have no sequential nature, and the convolution introduces unnecessary bias.
>
> - (b) Datasets with channel order (e.g., PEMS):
> The performance remains **nearly identical** even after removing 1D-conv. We acknowledge the reviewer’s expectation that removing 1D-conv from datasets **with** channel order should **degrade** performance. However, the **lack of degradation** suggests that the 1D-conv **did not effectively capture** the sequential structure in PEMS, rather than implying that such structure does not exist. In fact, the stable performance further **supports the rationale for its removal**, indicating that FSMamba generalizes well regardless of whether the channel order is explicitly present.
>
>
> Overall, these findings **strengthen** the motivation for removing 1D-conv and demonstrate robustness across **both (few) ordered and (general) unordered datasets**.
>
>
> &nbsp;
>
> Notably, PEMS is an ***exception*** where channels have a sequential order, unlike the general TS datasets.
> The PEMS datasets possess a meaningful geographical order, and our paper ***explicitly treats them as an “exception”***, not as the norm. As stated in **Table 9 caption** and **L358–360**, *“PEMS is an exception where channels follow a geographical order which can be known in advance via metadata.”* Our motivation for removing the 1D-conv primarily concerns **”general”** TS datasets (e.g., ETT, Weather), where the channel dimension, by nature, **lacks the sequential characteristics inherent to the temporal dimension**.
>
>
> &nbsp;
>
> ---
> &nbsp;
>
> > [Q2] Does the conclusion in Fig. 7 and the robustness generalize to even **higher-dimensional data** [1]?
>
>
> The **Traffic** dataset used in our experiments already includes **”862 channels”**, which is generally considered a **high-dimensional** multivariate TS benchmark. Note that the paper mentioned by the reviewer (*U-Cast: Learning Hierarchical Structures for High-Dimensional Time Series Forecasting* [A]) was released on **arXiv after our submission (2025-09-24)** and specifically addresses the newly proposed High-Dimensional Time Series Forecasting (HDTSF) setting.
>
>
> Nevertheless, following the reviewer’s feedback, we have **added experiments on two additional high-dimensional datasets** — **M5 (3049 channels)** and **S&P 500 (1475 channels)** — to further evaluate our method’s effectiveness on high-dimensional datasets. The datasets statistics and the results are shown below:
>
>
> &nbsp;
>
> | **Settings** | **M5** | **S&P 500** |
> |:-------------:|:------:|:------------:|
> | $L$ | 96 | 21 |
> | $H$ | 28 | 7 |
> | $C$ | **3,049** | **1,475** |
> | Frequency | Daily | Daily |
>
> &nbsp;
>
> | **Dataset** | **S-Mamba** |      | **FSMamba** |      |
> |:-------------|:-----------:|:----:|:------------:|:----:|
> |              | MSE | MAE | MSE | MAE |
> | **M5 Forecasting** | 0.374 | 0.872 | **0.372** | **0.865** |
> | **S&P 500 Index** | 0.399 | 0.267 | **0.395** | **0.265** |
>
>
> These results are now included in the revised version (**[Appendix L] Application to High-dimensional datasets**). If there are any additional datasets or aspects you would like us to further examine, please feel free to let us know!

---

> ### Author Response · Authors · 2025-11-21
>
> > [Q3] For the CSM pretraining task, why only preserving the **linear correlations**? Will preserving **non-linear correlations** also help?
>
>
> We appreciate the reviewer’s insightful comment. As noted in **L231–234**, we are ***NOT RESTRICTED TO A SPECIFIC METRIC*** such as linear correlation; correlation is merely one representative example. We employed correlation for the main experiments since it is **widely used in prior works [A,B] as a simple yet effective way to measure inter-channel relationships**, as noted in **L231–234**.  Moreover, as shown in **Table 10**, our method has been validated using **five different similarity metrics**, confirming that the approach is not tied to a particular form of correlation.
>
>
> We are open to incorporating additional metrics that the reviewer has in mind, including those capturing non-linear dependencies, and will gladly extend the experiments accordingly!
>
> &nbsp;
>
>
> - [A] Yingnan Yang, Qingling Zhu, and Jianyong Chen. Vcformer: Variable correlation transformer with inherent lagged correlation for multivariate time series forecasting. arXiv preprint arXiv:2405.11470, 2024.
> - [B] Lifan Zhao and Yanyan Shen. Rethinking channel dependence for multivariate time series forecasting: Learning from leading indicators. In ICLR, 2024.
>
>
> &nbsp;
>
> ---
> &nbsp;
>
> > [Q4] Can the **two loss strategies** (L_reg and L_CSM) be applied to other CD models like iTransformer[2] and Duet[3]?
>
>
> - **[$L_{\text{reg}}$]** The $L_{\text{reg}}$ loss is specifically designed to address the **bias issue inherent in Mamba’s sequential scanning mechanism**, and thus it is intended for **Mamba-based** TS forecasting models that capture CD, rather than for all CD models in general.
> - **[$L_{\text{CSM}}$]** The $L_{\text{CSM}}$ loss can be applied **not only to Mamba-based architectures** but also to other CD models such as Transformer-based ones.  While we initially validated it on two Mamba-based models (as shown in **Table 11**), we have also conducted an **additional experiment** applying CSM to iTransformer [A], a representative Transformer-based approach for capturing CD. The results are shown below with three different datasets with various number of channels (ETTh1:7, Weather:21, ECL:321), demonstrating the **effectiveness of CSM beyond Mamba-based models**.
>
>
> |   | **ETTh1** |  |  |  |    **Weather** |   |  |     | **ECL** |    | |      |
> |:--|:--|:--:|:--:|:--:|:--|:--:|:--:|:--:|:--|:--:|:--:|:--:|
> |     | MSE | MAE | MSE | MAE | MSE | MAE | MSE | MAE | MSE | MAE | MSE | MAE |
> | $H$    |  -  |  -   | +CSM | +CSM    |  -  |   - | +CSM | +CSM    |  -  | -   | +CSM |  +CSM   |
> | **96**  | .387 | .405 | **.385** | **.401** | .174 | .215 | **.174** | **.210** | .148 | .240 | **.145** | **.237** |
> | **192** | .441 | .436 | **.441** | **.436** | .224 | .258 | **.221** | **.255** | .167 | .258 | **.163** | **.252** |
> | **336** | .487 | .458 | **.484** | **.453** | .281 | .298 | **.280** | **.296** | .179 | .272 | **.174** | **.266** |
> | **720** | .509 | .494 | **.491** | **.487** | .359 | .351 | **.357** | **.348** | .220 | .310 | **.215** | **.304** |
>
>
> These results are now included in the revised version (**[Appendix K] Application of CSM to iTransformer**). If there are any additional datasets or aspects you would like us to further examine, please feel free to let us know!
>
> &nbsp;
>
> - [A] Liu Y, Hu T, Zhang H, Wu H, Wang S, Ma L, Long M. itransformer: Inverted transformers are effective for time series forecasting. ICLR 2024.
>
> &nbsp;
>
>
> **Please let us know if there are any remaining issues you'd like to discuss!**

---

### Official Review · Reviewer_6NWD · 2025-11-02

**Soundness:** 3
**Presentation:** 3
**Contribution:** 3
**Rating:** 6
**Confidence:** 4

**Summary:**

This paper investigates whether bidirectionality is essential in applying Mamba for modeling channel dependencies (CD) in multivariate time series forecasting. The authors propose FSMamba, a lightweight alternative that removes bidirectionality and introduces a regularization term to align embeddings generated from original and reversed channel orders. Experiments on thirteen benchmark datasets (including ETT, PEMS, Exchange, Weather, ECL, Solar, and Traffic) demonstrate that FSMamba achieves state-of-the-art accuracy with around 37% fewer parameters than prior Mamba-based models and exhibits enhanced robustness to channel-order permutations.

**Strengths:**

1. The paper asserts that minimizing embedding distance improves robustness to channel order but does not formally analyze why this suffices to approximate bidirectional behavior. A stronger theoretical connection between the regularizer and bidirectionality could strengthen the contribution.
2. The paper only reports λ-sensitivity results on the ETTh1 dataset (Table 14), showing stable performance within [0.01, 0.1], but it does not provide quantitative evidence across other datasets such as Weather, ECL, or PEMS. As the regularization term is central to FSMamba’s robustness design, a more systematic analysis of λ’s influence across datasets would strengthen the reliability and generality of the proposed approach.

**Weaknesses:**

1. The paper asserts that minimizing embedding distance improves robustness to channel order but does not formally analyze why this suffices to approximate bidirectional behavior. A stronger theoretical connection between the regularizer and bidirectionality could strengthen the contribution.
2. The paper only reports λ-sensitivity results on the ETTh1 dataset (Table 14), showing stable performance within [0.01, 0.1], but it does not provide quantitative evidence across other datasets such as Weather, ECL, or PEMS. As the regularization term is central to FSMamba’s robustness design, a more systematic analysis of λ’s influence across datasets would strengthen the reliability and generality of the proposed approach.

**Questions:**

1.Could you provide a more formal justification of how the proposed regularization approximates bidirectional scanning?
2.How sensitive is FSMamba to the regularization weight λ across datasets beyond ETTh1?

---

> ### Author Response · Authors · 2025-11-21
>
> > [W1] The paper asserts that minimizing embedding distance improves robustness to channel order but **does not formally analyze why this suffices to approximate bidirectional behavior**. A stronger theoretical connection between the regularizer and bidirectionality could strengthen the contribution.
>
>
> > [Q1] Could you provide a more formal justification of **how the proposed regularization approximates bidirectional scanning**?
>
>
>
>
> We emphasize that our goal is **NOT to directly approximate bidirectional behavior**, but rather to **replace the “bidirectional“ design with a "regularization"-based approach** that addresses this bias more effectively. As shown in **Table 1** and **Tables 6 and 8**, the bidirectional Mamba fails to handle the bias, while the proposed regularization effectively mitigates it, demonstrating superior effectiveness. By aligning representations across scan directions, the proposed regularization mitigates the bias, rendering the distinction between **unidirectional** and **bidirectional** architectures largely insignificant.
>
>
>
>
>
>
>
>
> Our goal is not to replicate bidirectional behavior but to challenge the necessity of the bidirectional design. While it was intended to address sequential order bias, our regularization fulfills this role more effectively by aligning representations across directions—making the distinction between uni- and bidirectional architectures largely insignificant once the bias is mitigated.
>
>
> The **sequential order bias** arises from applying Mamba’s sequential scan to the channel dimension where no inherent order exists, and our method mitigates this through **”regularization”** rather than architectural modification (e.g., **”bidirectional design”**).
>
>
> Moreover, we empirically demonstrate the effectiveness of regularization in handling the bias through multiple experiments as below:
> - 1. [**Figure 8**] Visualization of forward and reversed embeddings
>   - Forward and reversed embeddings converge to nearly identical manifolds.
> - 2. [**Table 8**] Effectiveness or regularization
>   - The results confirm that regularization effectively reduces order bias and enhances the stability of both uni- and bidirectional Mamba models.
> - 3. [**Table 3**] Main results on TS forecasting
>   - FSMamba achieves superior performance compared to bidirectional models (e.g., S-Mamba).
> - 4. [**Table 12**] Channel-order robustness
>   - FSMamba remains robust under randomly permuted channel orders.
>
>
> Overall, these results demonstrate that the **proposed regularization effectively fulfills the intended role of the bidirectional design**, offering a simpler yet empirically validated solution to sequential order bias.
>
>
>
>
> &nbsp;
>
> ---
> &nbsp;
>
> > [W2] The paper only reports λ-sensitivity results on the ETTh1 dataset (Table 14), showing stable performance within [0.01, 0.1], but it does not provide quantitative evidence across other datasets such as Weather, ECL, or PEMS. As the regularization term is central to FSMamba’s robustness design, a more systematic analysis of λ’s influence across datasets would strengthen the reliability and generality of the proposed approach.
>
>
> > [Q2] How sensitive is FSMamba to the regularization weight λ across datasets beyond ETTh1?
>
>
> We would like to clarify that **Table 14** reports results not only for ETTh1 but for **all four ETT datasets (ETTh1, ETTh2, ETTm1, ETTm2)**, each averaged over **four forecasting horizons**—resulting in 16 total settings × 6 λ values, which we believe already covers a broad range of conditions.
>
>
> Nevertheless, to further support the reviewer’s concern regarding generality, we conducted additional experiments on **Weather** and **ECL** datasets, with the results summarized below:
>
>
> | Dataset | w/o reg. (λ=0) | λ=0.0001 | λ=0.001 | λ=0.01 | λ=0.1 | λ=0.2 | λ=0.5 | S-Mamba |
> |:--|:--:|:--:|:--:|:--:|:--:|:--:|:--:|:--:|
> | ETTh1 | .455 | .437 | .434 | .434 | .434 | .434 | .434 | .457 |
> | ETTh2 | .383 | .380 | .378 | .378 | .378 | .378 | .378 | .383 |
> | ETTm1 | .403 | .394 | .392 | .392 | .392 | .392 | .393 | .398 |
> | ETTm2 | .289 | .283 | .281 | .281 | .281 | .281 | .282 | .290 |
> | **Weather** | .253 | **.248** | **.247** | **.246** | **.246** | **.246** | **.246** | .252 |
> | **ECL** | .174 | .**166** | **.164** | **.163** | **.163** | **.163** | **.165** | .174 |
>
>
> The results consistently demonstrate that FSMamba **remains robust to the choice of lambdaacross diverse datasets**.
>
> &nbsp;
>
> **Please let us know if there are any remaining issues you'd like to discuss!**

---

### Meta-Review · Area_Chair_ypdx · 2026-01-13

**Summary:**

This paper investigates whether bidirectionality is essential in applying Mamba for modeling channel dependencies (CD) in multivariate time series forecasting. The authors have shown that unidirectional Mamba-style model, paired with a flipped-siamese regularizer, can match or surpass bidirectional Mamba baselines for multivariate forecasting. The paper is generally well written and includes extensive experiments and ablations across 13 datasets. However, the key technical justification is weak as it does not convincingly explain why embedding alignment under a single reverse-order constraint should substitute for bidirectionality or yield meaningful channel-order invariance, and the claimed robustness is limited. In addition, key methodological choices, e.g., applying Mamba to channel dependencies, the necessity and novelty of CSM pretraining, are not sufficiently motivated or justified. Therefore, I recommend reject.

**Reviewer Concerns:**

The concerns over high-dimensional generalization, non-linear similarity flexibility, transferability of losses, and the motivation for Mamba-on-CD have been addressed. However, the authors does not convincingly explain why embedding alignment under a single reverse-order constraint should substitute for bidirectionality or yield meaningful channel-order invariance, and the claimed robustness seem limited. The necessity and novelty of CSM pretraining, are also not sufficiently motivated or justified even after the rebuttal.

**Reviewer Scores:**

No reviewer seem would update the score.

---

### Decision · Program_Chairs · 2026-01-26

Reject